# SuffixDecoding: Extreme Speculative Decoding for Emerging AI Applications

**Gabriele Oliaro**[§,†]    **Zhihao Jia**[†]    **Daniel Campos**[§]    **Aurick Qiao**[§]

[§]Snowflake AI Research    [†]Carnegie Mellon University

{goliaro,zhihaoj2}@cs.cmu.edu,
{daniel.campos,aurick.qiao}@snowflake.com

⌂ **Project Page:** https://suffix-decoding.github.io
 **Code:** https://github.com/snowflakedb/ArcticInference

## Abstract

Speculative decoding is widely adopted to reduce latency in large language model (LLM) inference by leveraging smaller draft models capable of handling diverse user tasks. However, emerging AI applications, such as LLM-based agents, present unique workload characteristics: instead of diverse independent requests, agentic frameworks typically submit repetitive inference requests, such as multi-agent pipelines performing similar subtasks or self-refinement loops iteratively enhancing outputs. These workloads result in long and highly predictable sequences, which current speculative decoding methods do not effectively exploit. To address this gap, we introduce *SuffixDecoding*, a novel method that utilizes efficient suffix trees to cache long token sequences from prompts and previous outputs. By adaptively speculating more tokens when acceptance likelihood is high and fewer when it is low, SuffixDecoding effectively exploits opportunities for longer speculations while conserving computation when those opportunities are limited. Evaluations on agentic benchmarks, including SWE-Bench and Text-to-SQL, demonstrate that SuffixDecoding achieves speedups of up to $5.3\times$, outperforming state-of-the-art methods—$2.8\times$ faster than model-based approaches like EAGLE-2/3 and $1.9\times$ faster than model-free approaches such as Token Recycling. SuffixDecoding is open-sourced at https://github.com/snowflakedb/ArcticInference.

## 1 Introduction

Large language models (LLMs) are foundational to agentic AI applications, such as automated coding assistants [Wang et al., 2025, Yang et al., 2024], multi-agent workflows [Wang et al., 2024a, Chen et al., 2024a], and retrieval systems [Wang et al., 2024d, Gao et al., 2024b]. Unlike basic chatbots, these workloads issue repetitive and predictable inference requests. For instance, multi-agent systems repeatedly perform similar tasks, and reasoning loops [Wang et al., 2023a, Madaan et al., 2023] regenerate similar token sequences. Despite this predictable repetition, existing methods fail to fully exploit recurring patterns, leaving latency as a bottleneck.

A popular strategy for mitigating inference latency is *speculative decoding* [Leviathan et al., 2023, Chen et al., 2023, Miao et al., 2024, Cai et al., 2024, Lin et al., 2024, Zhang et al., 2024]. While an LLM can only generate one token per forward pass, it can verify *multiple* tokens. Leveraging this phenomenon, speculative decoding methods use small "draft" models or additional decoding heads to predict multiple candidate tokens, which the LLM then verifies in parallel.

To efficiently handle the long repetitions common in agent-driven applications, speculative decoding methods must satisfy two critical requirements. First, they need to generate draft tokens rapidly and

with minimal overhead, enabling maximal exploitation of long speculation lengths. Second, they must do so *adaptively*—only generating more draft tokens when acceptance likelihood is high and fewer tokens when acceptance likelihood is low, to prevent verification from becoming a bottleneck.

However, existing speculative decoding approaches fall short in meeting these dual requirements. Model-based methods can use significant GPU time when speculating long sequences, and can incur memory contention and kernel-level transitions [Chen et al., 2024b, Li et al., 2024] that must be managed carefully. Conversely, existing model-free approaches, such as prompt-lookup decoding (PLD) [Saxena, 2023], achieve low overhead and rapid token generation, but typically lack adaptivity. These methods speculate a fixed number of tokens irrespective of acceptance likelihood, leading to wasted computational resources on verifying long and improbable draft sequences.

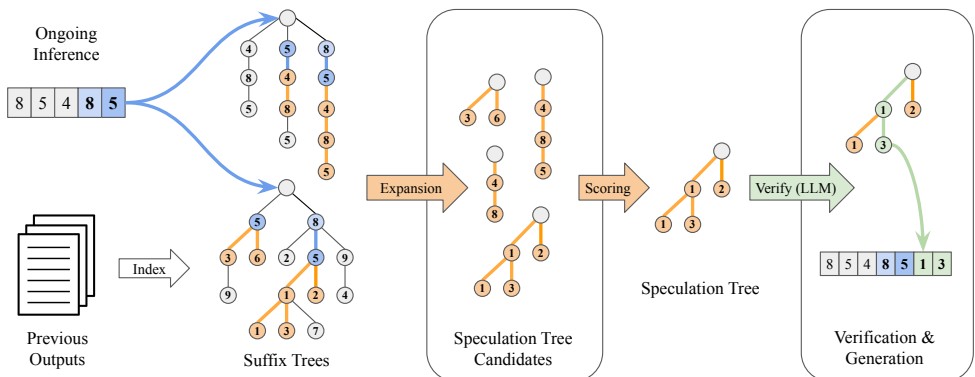

Figure 1: Overview of SuffixDecoding's algorithm. Two suffix trees track ongoing inference (top-left) and previous outputs (bottom-left). SuffixDecoding uses these trees to find matching patterns based on recently generated tokens. It constructs a speculation tree (middle) by selecting the most likely continuations, scoring them based on frequency statistics. Finally, the best candidate is verified by the LLM in a single forward pass (right), with accepted tokens (shown in green) being added to the output and used for the next round of speculation.

To address these limitations, we introduce *SuffixDecoding* (Fig 1), a model-free speculative decoding method for repetitive, agent-driven workloads. SuffixDecoding uses efficient suffix trees to cache long token sequences from prompts and previous outputs. Each node represents a token, and paths encode previously observed subsequences, enabling rapid pattern matching to identify continuations based on prior occurrences. Draft tokens are generated extremely quickly (∼20 microseconds per token) without GPU overhead.

At each inference step, SuffixDecoding adaptively limits its number of draft tokens based on the length of the pattern match, and uses frequency-based statistics captured within the suffix trees to score and select the best speculation candidate. Longer pattern matches enable confident speculation of longer token sequences, maximizing its effectiveness on agentic workloads, while shorter pattern matches trigger conservative speculation to avoid computational waste. Moreover, SuffixDecoding can seamlessly integrate with existing model-based speculative decoding methods. This flexibility enables a hybrid approach that leverages suffix-tree-based speculation for repetitive, predictable agentic workloads, while exploiting the strengths of model-based speculation methods for open-ended conversational tasks, thus achieving the best of both worlds.

We evaluate SuffixDecoding on two practical agent-driven workloads: SWE-Bench, an LLM-based software engineering benchmark, and AgenticSQL, a proprietary multi-agent pipeline application for SQL generation. We compare with state-of-the-art model-based and model-free speculative decoding methods using Spec-Bench [Xia et al., 2024], showing up to 2.8× faster decoding than EAGLE-2/3 [Li et al., 2025a], and 1.9× faster decoding than Token Recycling [Luo et al., 2024]. For SWE-Bench, we also measured the comprehensive, end-to-end task completion time—including prompt prefilling, token generation, and execution of external actions—and demonstrate speculative speedups of up to 4.5×. These results highlight that SuffixDecoding substantially reduces latency for real-world agentic applications, addressing a critical bottleneck in practical inference scenarios.

## 2 Background and Related Work

**LLM Inference.** LLM inference involves two stages: given a prompt $x_{\text{prompt}} = (x_1, x_2, \ldots, x_m)$, the LLM first processes the prompt in parallel (prefill), then sequentially generates new tokens (decode), with each token $x_{t>m}$ conditioned on previously generated tokens:

$$x_{t+1} = Sample(x|x_{1,\ldots,t}).$$

In greedy sampling, the highest-probability token is selected iteratively until a stopping condition. Since each token depends on preceding outputs, generation is inherently sequential, requiring a separate forward pass per token, limiting throughput and underutilizing parallel hardware accelerators.

**Speculative Decoding.** Speculative decoding [Leviathan et al., 2023] accelerates inference by generating multiple candidate tokens quickly using a lightweight model, which can then be verified in parallel by the primary LLM. The basic method has two core steps:

1. *Speculation*: A smaller "draft" model rapidly produces speculative tokens $x_{\text{spec}} = (x_{t+1}, \ldots, x_{t+n})$ based on the existing token prefix $x_{<t}$.
2. *Verification*: The LLM verifies the draft tokens in parallel, accepting tokens up to the first discrepancy and discarding the rest.

This shifts computation from sequential generation to parallel verification. However, draft models still require compute resources and add orchestration complexity. Model-based methods include Medusa [Cai et al., 2024], SpecInfer [Miao et al., 2024] (which introduced tree-based speculation), multi-token prediction [Gloeckle et al., 2024], and blockwise parallel decoding [Stern et al., 2018, Kim et al., 2024].

Recent *model-free* methods include Prompt Lookup Decoding [Saxena, 2023], Token Recycling [Luo et al., 2024], LLMA [Yang et al., 2023], and ANPD [Ou et al., 2024]. These rely on small reference texts and lack adaptive speculation. SuffixDecoding is uniquely designed for agentic applications with long repetitive sequences.

**Agentic AI Algorithms.** Agentic applications structure tasks as multiple LLM calls, generating long repetitive token sequences. *Self-consistency [Wang et al., 2023a]* samples multiple reasoning paths from the same prompt, sharing similar chain-of-thought steps. *Self-refinement [Madaan et al., 2023]* iteratively fixes errors, revising small portions while preserving surrounding content. *Multi-agent workflows [Khot et al., 2023]* use specialized agents with narrow functions, producing repetitive outputs. These patterns create opportunities for exploiting long repeated sequences.

**Methods for Accelerating LLM Agents.** Recent works target latency in agentic applications. ALTO [Santhanam et al., 2024] optimizes multi-agent workflows through pipelining and scheduling. Dynasor [Fu et al., 2024] early-terminates unlikely reasoning paths. SuffixDecoding takes an orthogonal speculative decoding approach and can be used in combination.

## 3 SuffixDecoding

The goal of *SuffixDecoding* is to enable fast, adaptive speculative decoding over long sequences, particularly suited for agentic applications where repeated inference calls often contain highly predictable and overlapping token sequences. In such settings, long stretches of output can be accurately predicted from prior and ongoing requests.

To fully exploit these opportunities, SuffixDecoding must address two key challenges. First, it must support *fast generation of speculative sequences*—including long continuations—without relying on draft models or expensive token-by-token prediction. Second, it must be *adaptive* to the current prediction context: aggressively speculating long continuations only when they are likely to be accepted, and speculating shorter sequences when uncertain to avoid wasted verification compute.

To support fast speculation, SuffixDecoding builds a *suffix tree* [Weiner, 1973] over the tokens in the current and prior requests, and uses the suffix tree to generate speculative tokens. The root node of the tree represents the beginning of a suffix of any token sequence stored in the tree, each child of

a node represents a specific token which is a possible continuation from its that node, and the path from the root to each node represents a distinct subsequence.

For each request, we consider speculating token sequences from (1) the prompt and output of that request, and (2) the outputs of prior requests. Doing so captures the source of output token repetition from many agentic algorithms, including self-consistency, self-refinement, and multi-agent pipelines.

SuffixDecoding leverages suffix trees to perform fast pattern matching and find possible continuations of token sequences. Suppose the prompt and output tokens of an ongoing inference is $x_{1:t}$. Consider a suffix $x_{t-p+1:t}$ of length $p$, which we will refer to as the *pattern* sequence. We walk the suffix tree starting from the root node, and at each step taking the child that corresponds to token $x_{t-p+i}$. If no such child exists, then the pattern is not found and SuffixDecoding reverts to standard non-speculative decoding. Otherwise, after $p$ steps, we arrive at a node whose descending paths are the possible continuations of the pattern sequence.

Although this procedure can quickly find a (potentially large) set of candidate sequences, verifying all of them in speculative decoding may be cost-prohibitive. Instead, SuffixDecoding builds a much smaller and more likely speculation tree through a greedy expansion and scoring procedure, and uses this smaller tree in tree-based speculation. An overall illustration of SuffixDecoding is shown in Fig. 1, which we detail in the rest of this section.

**Suffix Tree Construction.** Building the suffix tree and updating it as part of an online inference service involves two stages. First, the previous inference outputs can be added to the tree in a single offline processing step (e.g. from historical logs), or online during inference serving after each inference request completes. Second, the current ongoing prompt and out tokens are added online as new requests are received and as each new token is generated.

In reality, we found it convenient to maintain two different suffix trees: a *global* tree for the previously generated outputs, and a separate *per-request* tree for the current ongoing inference request. This circumvents the complexities and overheads due to synchronizing the suffix tree updates from multiple concurrent requests. The global tree can be constructed offline in $O(n)$ time, while the per-request tree can be efficiently constructed and updated online [Ukkonen, 1995].

Although suffix trees are memory-efficient at $O(n)$ space, the global tree can still become large when there are many previous outputs. However, they only require CPU memory, which is typically plentiful and under-utilized in LLM serving scenarios. For example, AWS `p5.48xlarge` are often used for LLM serving and have 2TB of main memory, which is easily enough to support a suffix tree over millions of historical outputs and billions of tokens. Given typical server configurations, SuffixDecoding can cache approximately a month's worth of generated tokens before requiring cache eviction (see Appendix B for detailed memory overhead analysis).

**Speculation Tree Expansion.** Given a pattern sequence $x_{t-p+1:t}$ of an ongoing inference $x_{1:t}$, SuffixDecoding can quickly find a node $N_p$ in the global or per-request suffix tree whose descending paths are the possible continuations of the pattern sequence. To select a smaller more likely sub-tree that is of a more practical size for speculative verification, we start with the single node $N_p$ and grow a sub-tree greedily by expanding one leaf node at a time.

In particular, we define:

$$C(N) = \frac{\texttt{COUNT}(N)}{\sum_{M \in \texttt{CHILDREN}(\texttt{PARENT}(N))} \texttt{COUNT}(M)}$$

$$D(N) = \begin{cases} D(\texttt{PARENT}(N)) \times C(N), & \text{if } N \neq N_p \\ 1, & \text{otherwise} \end{cases},$$

where $\texttt{COUNT}(N)$ is the number of occurrences of node $N$ in the reference corpus, which can be computed when constructing the suffix tree. Starting with the single node $N_p$ in our speculation sub-tree, we consider all children of all of its leaf nodes, and add the node $N$ with the highest $D(N)$. This process is repeated until the sub-tree reaches a predetermined size limit, `MAX_SPEC`.

Intuitively, $C(N)$ estimates the probability that $\texttt{TOKEN}(N)$ would be the next observed token in a sub-sequence $\texttt{TOKEN}(N_p), \ldots, \texttt{TOKEN}(\texttt{PARENT}(N))$, and $D(N)$ estimates the probability that $\texttt{TOKEN}(N)$ would be ultimately accepted by the speculative tree verification, assuming the output tokens follow

historical patterns. Thus, SuffixDecoding builds the speculation tree by greedily adding leaf nodes that it believes to be the most likely to be accepted during verification.

---

**Algorithm 1** Speculation Tree Generation

---

**function** EXPANDSPECULATIONTREE(N_p, MAX_SPEC)
 **Input:** Suffix tree node $N_p$, MAX_SPEC
 Initialize $T \leftarrow \{N_p\}$
 **while** $|T| <$ MAX_SPEC **do**
  $N \leftarrow \arg\max_{N \in \text{CHILDREN(LEAVES}(T))} D(N)$
  $T \leftarrow T \cup \{N\}$
 **end while**
 **return** $T$
**end function**
**function** MATCHPATTERN(S, x_{1:t}, p)
 **Input:** Suffix tree $S$, sequence $x_{1:t}$, length $p$
 Initialize $N_p \leftarrow \text{ROOT}(S)$
 **for** $i = 1$ **to** $p$ **do**
  **if** NO_CHILD$(N_p, x_{t-p+i})$ **then**
   **return** $\emptyset$
  **end if**
  $N_p \leftarrow \text{CHILD}(N_p, x_{t-p+i})$
 **end for**
 **return** $N_p$
**end function**
**function** GENERATECANDIDATETREE(S_g, S_r, x_{1:t}, $\alpha$, P)
 **Input:** Global suffix tree $S_g$, prompt suffix tree $S_r$, sequence $x_{1:t}$, max spec factor $\alpha$, max pattern size $P$
 Initialize $T_{\text{best}} \leftarrow \emptyset$, $\text{SCORE}_{\text{best}} \leftarrow 0$
 **for** $S$ **in** $\{S_g, S_r\}$ **do**
  **for** $p = 1$ **to** $P$ **do**
   $N \leftarrow \text{MatchPattern}(S, x_{1:t}, p)$
   $T \leftarrow \text{ExpandSpeculationTree}(N, \alpha p)$
   **if** $\text{SCORE}(T) > \text{SCORE}_{\text{best}}$ **then**
    $T_{\text{best}} \leftarrow T$
    $\text{SCORE}_{\text{best}} \leftarrow \text{SCORE}(T)$
   **end if**
  **end for**
 **end for**
 **return** $T_{\text{best}}$
**end function**

---

**Adaptive Speculation Lengths.** While the procedure above allows SuffixDecoding to cache and quickly speculate long token sequences based on empirical probability estimates, it also needs a mechanism for *adaptively* controlling the number of tokens it speculates. SuffixDecoding achieves this by dynamically adjusting `MAX_SPEC`. Low values mean fewer but more likely tokens would be chosen for speculation, while higher values mean more but less likely tokens would be chosen. If too low, then the speedup from speculation can be limited, and if too high, then compute may be wasted on verifying unlikely tokens.

To guide how to adaptively set `MAX_SPEC`, we observed that the number of accepted tokens in practice typically increases with longer pattern sequence lengths $p$ (Fig. 2a). Thus, we define `MAX_SPEC` adaptively as

$$\texttt{MAX\_SPEC}(p) = \alpha p,$$

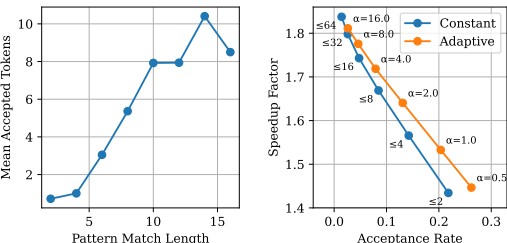

(a) Avg accepted tokens vs pattern match length.

(b) Using constant vs adaptive `MAX_SPEC`.

Figure 2: (a) the mean number of accepted tokens increases with the length of the pattern match, which motivates `MAX_SPEC` $= \alpha p$. (b) shows that this choice achieves a better trade-off between acceptance rate and speculative speedup.

where $\alpha$ is a user-defined max speculation factor. Fig. 2b shows that setting `MAX_SPEC` adaptively according to the pattern length results in a better trade-off between acceptance rate and speculative speedup. In practice, we found that $\alpha \in [1, 4]$ works well for agentic applications.

**Speculation Tree Scoring.** So far, we have discussed how to obtain a speculation tree given a suffix tree and a pattern length $p$. However, SuffixDecoding maintains two suffix trees, the global suffix tree and the per-request suffix tree, each with many choices for $p$. To obtain just a single speculation tree, we build speculation trees for both the global suffix tree and the per-request suffix tree, and for a range of values of $p$. Then, a single speculation tree is selected according to a scoring function:

$$\text{SCORE}(T_{spec}) = \sum_{N \in T_{spec}} D(N).$$

Intuitively, if $D(N)$ estimates the probability that node $N$ in a speculation tree $T_{spec}$ would be accepted, then $\text{SCORE}(T_{spec})$ estimates the expected number of accepted tokens. SuffixDecoding then selects the $T_{spec}$ with the highest `SCORE` as the final speculation tree to be verified. The end-to-end candidate generation from speculation tree expansion to scoring is described in Alg. 1.

***Hybrid* Suffix Speculative Decoding.** Lastly, we find that $\text{SCORE}(T_{spec})$ can be used to dynamically decide between using SuffixDecoding or falling back to a model-based speculation method, which is useful for practical scenarios when the workload can be mixed between agentic and more diverse applications. Specifically, for each decoding iteration, we always speculate using SuffixDecoding first. If $\text{SCORE}(T_{spec}) > \tau$, where $\tau$ is a configurable threshold, then SuffixDecoding's draft tokens are used. Otherwise, we use a fall-back speculation method, such as EAGLE-3 [Li et al., 2025a].

As a practical guideline, we find that setting $\tau$ close to the mean accepted tokens of the fallback speculator works well for mixed workloads, while using SuffixDecoding alone ($\tau = 0$) is optimal for highly repetitive agentic tasks. A detailed sensitivity analysis of $\tau$ across different workload types is provided in Appendix B.1. For batched serving scenarios, SuffixDecoding can be integrated with existing batch-level speculation control methods (see Appendix C).

## 4 Evaluation

### 4.1 Evaluation Methodology

**Baseline Comparisons.** We compare with both model-based and model-free speculative decoding methods using Spec-Bench [Xia et al., 2024]. (1) *EAGLE-2* and *EAGLE-3* [Li et al., 2025a], state-of-the-art model-based speculators, (2), *Prompt-Lookup Decoding (PLD)* [Saxena, 2023], a simple model-free speculator based on ngram-matching, and (3) *Token Recycling* [Luo et al., 2024], a more recent model-free speculator that sources token sequences from both the prompt and previous outputs. EAGLE-3 and Token Recycling both leverage tree speculation [Miao et al., 2024].

**Datasets and Agentic Applications.** We evaluate on both agentic and non-agentic workloads. For agentic applications, we trace requests from two real applications: OpenHands [Wang et al., 2024c] on *SWE-Bench* [Jimenez et al., 2024] (a GitHub issue resolution benchmark), and *AgenticSQL*, a proprietary multi-agent SQL generation workflow (Fig. 3). For non-agentic workloads, we use Spec-Bench, consisting of open-ended, single-turn tasks across 13 categories, in-

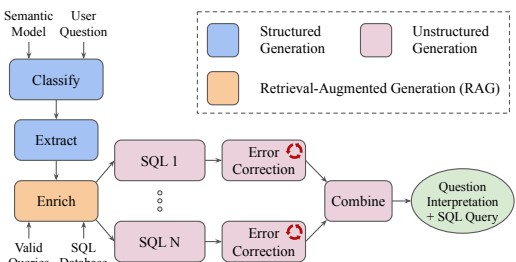

Figure 3: AgenticSQL is a multi-agent workflow consisting of stuctured generation, unstructured generation, and retrieval-augmented generation steps across several different LLMs. Useful features are extracted from the user question (Classify and Extract) and supplemented with retrieved context (Enrich). Several text-to-SQL steps propose solutions to the user question (SQL 1...N) in parallel with feedback from an error corrector. A last Combine step synthesizes the proposed SQL candidates into a final SQL query and text response.

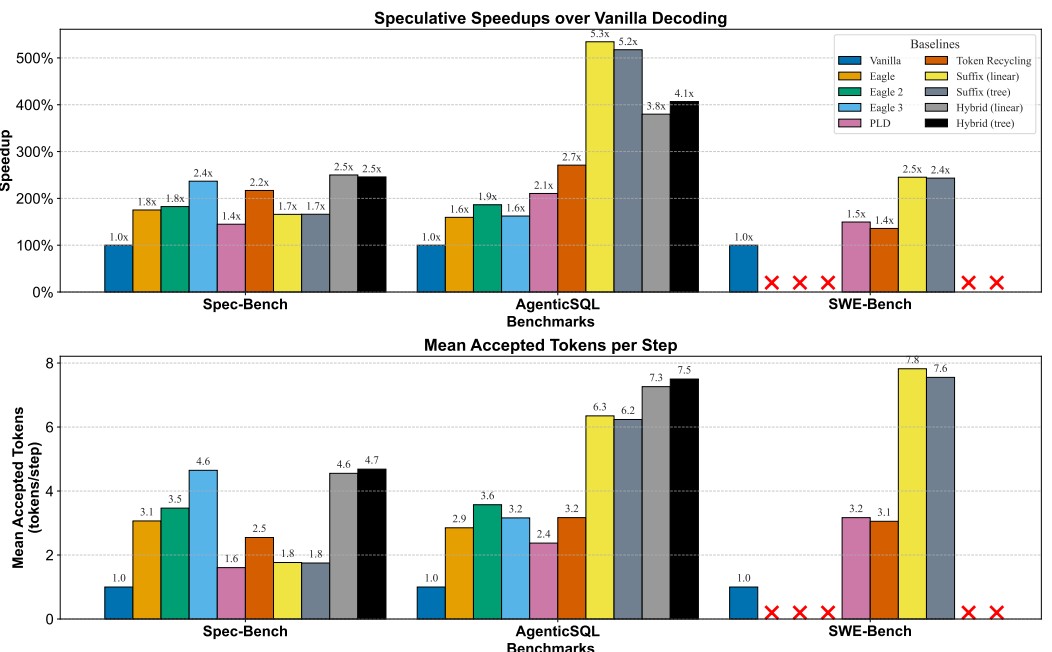

Figure 4: Speculative speedups (top) and mean accepted tokens per step (bottom) compared to vanilla decoding for SuffixDecoding and baseline methods on three benchmarks: Spec-Bench, AgenticSQL, and SWE-Bench. Experiments use Llama-3.1-8B-Instruct on a single H100 GPU with batch size 1. Speedup is measured as the ratio of wall-clock time-per-output-token relative to vanilla decoding. Suffix (tree) and Hybrid (tree) use SuffixDecoding's tree speculation algorithm, which constructs a speculation tree from the suffix tree for parallel verification. Suffix (linear) and Hybrid (linear) use a simpler linear speculation approach that only allows sequential token chains. The hybrid variants combine SuffixDecoding with EAGLE-3, dynamically selecting between suffix-based and model-based speculation based on pattern match confidence. Note that EAGLE-2/3 and Token Recycling failed to run on several SWE-Bench tasks due to long context lengths (>8192 tokens), indicated by missing bars. Spec-Bench represents a non-agentic workload and is included for comparison. Further sub-task breakdowns, including the raw time-per-output-token and mean acceptance lengths, can be found in Appendix A.1.

cluding 8 MT-Bench categories (Writing, Role-
play, Reasoning, Math, Coding, Extraction, STEM, Humanities). We include Spec-Bench to stress-test SuffixDecoding's limitations and evaluate the hybrid fallback mechanism.

**End-to-end System Evaluation.**    We implemented SuffixDecoding in vLLM [Kwon et al., 2023]. By running OpenHands live, we show SuffixDecoding accelerates end-to-end task completion times, including prefill and code execution.

**Hardware configuration.**    We conducted our experiments on a single `p5.48xlarge` AWS instance equipped with $8\times$ NVIDIA H100 80G GPUs and 2TB of main memory.

**Simulated Ablations.**    In addition to our main evaluation using real hardware, we also leverage a simulated verifier for additional experiments in Sec. 4.4 and continued in Appendix A.2. Given a prompt $x_{1:n}$ and example ground-truth response $y_{n+1:t}$, we can accurately simulate speculative verification for greedy sampling by verifying that speculated token $x_{n+i} = y_{n+i}$.

## 4.2   Baseline Comparisons

We compare SuffixDecoding with EAGLE-2, EAGLE-3, PLD, and Token Recycling on SWE-Bench and AgenticSQL. We also run the Spec-Bench standard dataset, which is a more traditional non-agentic workload. Fig. 4 shows the results. First, on the agentic workloads, SuffixDecoding

outperforms all baselines. In AgenticSQL, SuffixDecoding obtains a mean speedup of $5.3\times$ over vanilla decoding, a $2.8\times$ improvement over EAGLE-2/3, and $1.9\times$ higher than Token Recycling. In SWE-Bench, EAGLE-2/3 fail due to their maximum sequence length limitations. SuffixDecoding obtains a mean speedup of $2.5\times$ over vanilla decoding, a $1.7\times$ improvement over PLD, the next best baseline. SuffixDecoding's superior performance in agentic workloads can be attributed to its consistently higher mean accepted tokens per decoding step. In AgenticSQL, SuffixDecoding reaches 6.3 mean accepted tokens per step—substantially higher than EAGLE-3 (3.6 tokens) and Token Recycling (3.2 tokens). In SWE-Bench, SuffixDecoding achieves 7.8 mean accepted tokens per step, while PLD only accepts 3.2 tokens per step on average.

On non-agentic workloads such as Spec-Bench (which includes open-ended single-turn tasks and 8 MT-Bench categories), SuffixDecoding alone is outperformed by EAGLE-2/3 and Token Recycling, as expected for less repetitive scenarios. However, the hybrid approach of SuffixDecoding + EAGLE-3 achieves the best of both worlds: we speculate with the faster SuffixDecoding method whenever possible and fall back to EAGLE-3 when the speculation score is too low. The Hybrid approach obtains a mean speedup of $2.5\times$ over vanilla decoding, outperforming the $2.4\times$ speedup from standalone EAGLE-3 and the $2.2\times$ speedup from Token Recycling.

The hybrid approach also performs well in AgenticSQL, achieving a $4.1\times$ speedup in the tree variant, significantly better than the $1.9\times$ speedup from standalone EAGLE-2/3 and the $2.7\times$ speedup from Token Recycling. These speedups are achieved thanks to the hybrid approach's impressive 7.5 mean accepted tokens per step, more than $2\times$ higher than EAGLE-2/3 and Token Recycling. SuffixDecoding has a slightly lower mean acceptance length of 6.3, but its much lower speculation cost and higher acceptance rate make it the winning solution in agentic tasks ($5.3\times$ average speedup compared to the $4.1\times$ speedup of the hybrid approach).

**A peek into a speculation tree.**
To gain some intuition into why Suf­fixDecoding performs so well for cer­tain tasks, we examine how it builds a speculation tree for the AgenticSQL Extract task. The outputs of the Ex­tract task have many characteristics in common. First, they are all JSON doc­uments following the same format and key names, with keys often appearing in the same order. Second, many of the features are discrete values, and in particular, boolean true/false values.

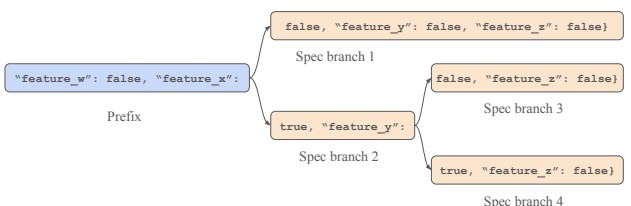

Figure 5: A SuffixDecoding speculation tree containing 66 tokens for the AgenticSQL Extract task.

These patterns are recorded in SuffixDecoding's global suffix tree and guide its speculation tree construction.

Fig. 5 shows an example of a speculation tree constructed by SuffixDecoding. We observed many instances of large speculation trees that branch at each boolean true/false value of several consecutive features. These speculation tree always contains a branch with high acceptance, advancing output generation by dozens of tokens or more in one step. Although this is one specific example of a speculation tree, it demonstrates that SuffixDecoding can find complex patterns in previous outputs, particularly for structured generation tasks, that help accelerate output generation.

## 4.3 End-to-End SWE-Bench on vLLM

In this section, we show that SuffixDecoding can be efficiently integrated into vLLM, a popular inference system used in production deployments, and it can effectively accelerate accelerate end-to-end agentic task completion time. For this experiment, we run OpenHands directly on vLLM with SuffixDecoding, so the agent is solving each benchmark problem live. We also use the specially-trained LLM `all-hands/openhands-lm-32b-v0.1-ep3`, which was fine-tuned for SWE-Bench and achieves 37.2% on SWE-Bench Verified. Since there are no model-based methods with draft models trained for this LLM, we compare with vLLM's native implementation of PLD.

Fig. 6 shows the results. First, we note that decoding time (i.e. output generation) takes a majority of the time across all SWE-Bench tasks, dominating both prefilling and agentic actions (i.e. code execution). In this end-to-end scenario, SuffixDecoding outperforms PLD by $1.3$–$3\times$, leading to a

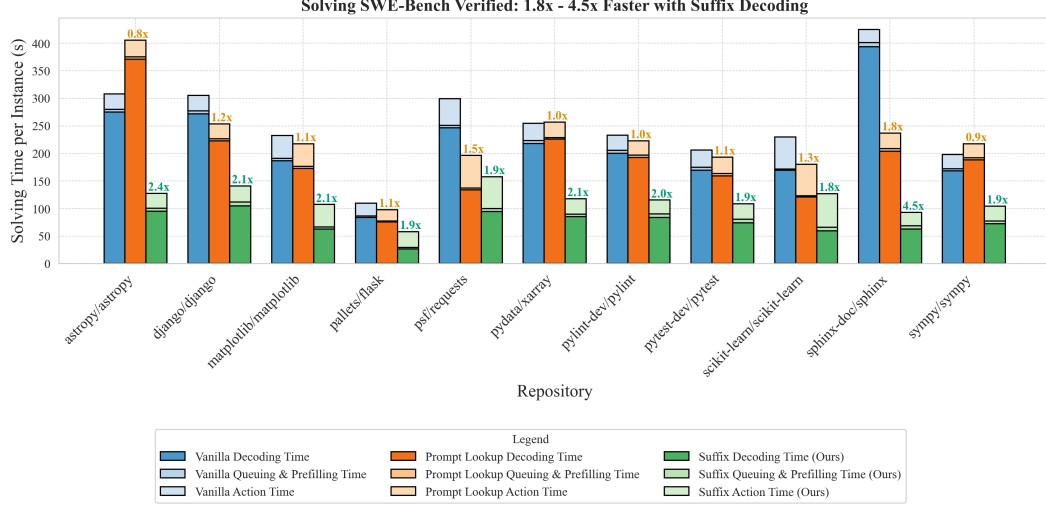

Figure 6: End-to-end task-completion time of the OpenHands agent on SWE-Bench Verified. The benchmarks are run with a concurrency of 8 tasks running simultaneously. vLLM is deployed on 4 H100 GPUs configured with 4-way tensor parallelism and prefix caching enabled. The results are broken down by the different code repositories in SWE-Bench.

1.8–4.5× speculative speedup over vanilla decoding. Since SuffixDecoding exactly preserves the output distribution of the LLM, it matches the original model's 37.2% score on SWE-Bench Verified.

## 4.4 Ablation Experiments

In this section, we present a few ablation studies on SuffixDecoding using a simulated verifier on offline traces. Given a ground-truth prompt-response pair from an LLM, we can verify the draft tokens proposed by SuffixDecoding by comparing with the ground truth responses. Additional ablation studies can be found in Appendix A.2.

**Global vs per-request suffix trees.** We study the impact of the two suffix trees: the global suffix tree containing previous outputs, and the per-request suffix tree containing the prompt and generation of the current ongoing request. To do so, we ran the tasks in AgenticSQL using SuffixDecoding (1) with the global suffix tree only, (2) with the per-request suffix tree only, and (3) using both trees.

Fig. 7 shows the results. First, we note that with the exception of the Content Enrichment (Enrich) and Extract steps, using both suffix trees performs better than using just one. The small degradations on the enrich and extract steps suggest that, when both trees are present, SuffixDecoding may sometimes choose a speculation tree from the per-request suffix tree when the global suffix tree may have been the better choice. Improvements to SuffixDecoding's speculation tree scoring mechanism may help bridge this gap.

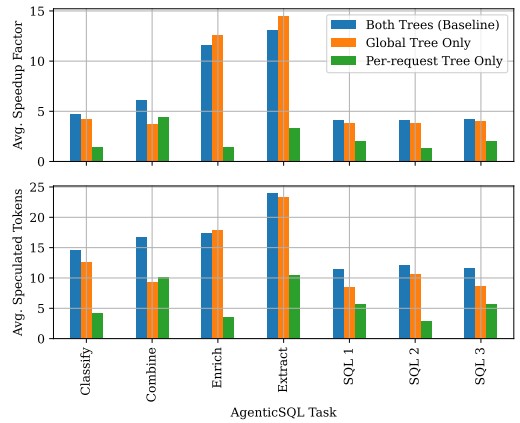

Figure 7: Speedup factor and number of speculated tokens for the tasks in AgenticSQL. SuffixDecoding was run with only the global suffix tree, only the per-request suffix tree, and both (baseline).

Second, the global tree outperforms the per-request tree on all tasks except for Combine. This is because the Combine task heavily re-uses tokens from its context, which are the proposed SQL

solutions from the previous steps in the workflow. Although there is a diversity of task characteristics, SuffixDecoding is able to achieve high speedups on all of them by combining both suffix trees.

**SuffixDecoding in open-ended scenarios.** Although SuffixDecoding is designed for agentic workloads with long repeated token sequences, it is also interesting to evaluate it using more open-ended workloads like WildChat (open-ended chat) [Zhao et al., 2024] and Magicoder (code-oriented chat) [Wei et al., 2023]. Details on these datasets can be found in Appendix A.2.

In Fig. 8, we show the speedup and acceptance rate of SuffixDecoding on WildChat and Magicoder across a range of suffix tree sizes between 256 and 10,000 output examples. First, we note a promising pattern: the speedup consistently improves as the size of the suffix tree grows. This indicates that SuffixDecoding can learn useful patterns even in workloads with lower token repetition, and may be a substitute for model-based methods when a draft model is not available.

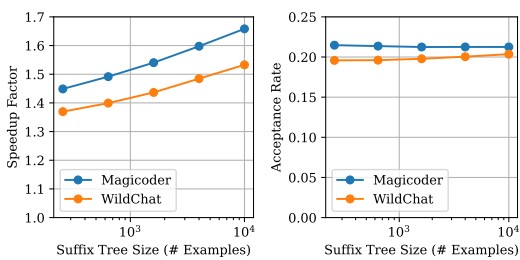

Figure 8: Speedup (left) and acceptance rate (right) vs global suffix tree size for Magicoder and Wildchat ($\alpha = 1$). The speedup from SuffixDecoding continues to increase with more previous output examples, while the acceptance rate holds steady.

Second, perhaps surprisingly, the acceptance rate does not change much even when the suffix tree size varies across almost two orders of magnitude. We believe this is primarily due to the effect of the adaptive speculation length $\texttt{MAX\_SPEC} = \alpha p$. Although less data may mean less certainty in the speculated tokens, the pattern matches are also shorter, which results in fewer speculated tokens.

## 5    Conclusion

In this paper, we presented SuffixDecoding, a model-free speculative decoding approach designed for emerging agentic applications. Using efficient suffix tree data structures, SuffixDecoding effectively exploits long and repetitive token sequences found in many agentic algorithms, such as self-consistency, self-refinement, and multi-agent pipelines. Using two practical agentic applications, OpenHands and AgenticSQL, we showed that SuffixDecoding significantly accelerates their decoding latency and task-completion times, and is also significantly faster than other model-based and model-free speculative decoding baselines.

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

# A Technical Appendices and Supplementary Material

## A.1 Details for Main Experiments

### A.1.1 Experiment setup details

**Setup of Spec-Bench experiments (Sec. 4.2).** We conducted our Spec-Bench experiments by running the Spec-Bench codebase from the original repository with the following modifications. First, we updated the code to work with the latest version of the `transformers` library, which is required to run recent open-source LLMs such as `meta-llama/Llama-3.1`. We also added support for arbitrary datasets (such as SWE-Bench and AgenticSQL) and implemented SuffixDecoding within the framework. We ran the experiments on a 8xH100 80GB GPU cluster, with 1TB RAM. We ran each baseline using one GPU, and a batch size of 1, just like in the original SpecBench code.

**Setup of vLLM SWE-Bench experiment (Sec. 4.3).** We conducted the end-to-end SWE-Bench experiment on a 8xH100 80GB GPU cluster, with 1TB RAM. We served the `all-hands/openhands-lm-32b-v0.1-ep3` model locally using vLLM, with a tensor parallelism degree of 4 and with prefix caching enabled. We used the flashinfer kernels for sampling. We made some minor modifications to vLLM to record the per-request and per-step statistics of interest (time-per-token latency, throughput, acceptance length, acceptance rate). We used the same settings for all baselines. We ran the OpenHands daemon on the same machine, and used the OpenAI API to interact with the vLLM server. We ran OpenHands with the CodeActAgent [Wang et al., 2024b] with `ITERATIVE_EVAL_MODE=true`, and a maximum of 100 iterations, as recommended by the OpenHands authors. We used a maximum of 16 concurrent workers to run the SWE-Bench tasks.

### A.1.2 Detailed sub-task results

The following tables present detailed performance metrics across all evaluation benchmarks. For Spec-Bench, the 13 categories include both single-turn open-ended tasks (qa, rag, math_reasoning, summarization, translation) and multi-turn tasks. Eight of these categories (Writing, Roleplay, Reasoning, Math, Coding, Extraction, STEM, Humanities) are from MT-Bench [Zheng et al., 2023], allowing direct performance comparison on this widely-used benchmark.

## SWE-Bench: Mean accepted tokens (tokens/step)

| System | astropy | django | matplotlib | seaborn | flask | requests | xarray | pylint | pytest | scikit-learn | sphinx | sympy | Overall |
|---|---|---|---|---|---|---|---|---|---|---|---|---|---|
| suffix (linear) | 6.415 | 6.546 | 5.521 | 7.207 | 3.772 | 6.480 | 7.137 | 4.635 | 5.412 | 5.933 | 17.140 | 5.165 | 7.821 |
| suffix (tree) | 6.262 | 6.221 | 4.992 | 7.064 | 3.708 | 5.922 | 6.764 | 4.452 | 5.311 | 5.600 | 16.876 | 5.020 | 7.552 |
| pld | 2.831 | 3.008 | 2.756 | 3.080 | 2.195 | 2.996 | 3.223 | 2.629 | 2.904 | 2.669 | 4.724 | 2.641 | 3.168 |
| recycling | 3.159 | 3.058 | 3.004 | 3.133 | - | - | 2.978 | 3.072 | 2.992 | 3.046 | 2.994 | - | 3.054 |
| vanilla | 1.000 | 1.000 | 1.000 | 1.000 | 1.000 | 1.000 | 1.000 | 1.000 | 1.000 | 1.000 | 1.000 | 1.000 | 1.000 |
| eagle3 | - | - | - | - | - | - | - | - | - | - | - | - | - |
| eagle2 | - | - | - | - | - | - | - | - | - | - | - | - | - |
| eagle | - | - | - | - | - | - | - | - | - | - | - | - | - |
| hybrid | - | - | - | - | - | - | - | - | - | - | - | - | - |

## SWE-Bench: Mean Acceptance Rate

| System | astropy | django | matplotlib | seaborn | flask | requests | xarray | pylint | pytest | scikit-learn | sphinx | sympy | Overall |
|---|---|---|---|---|---|---|---|---|---|---|---|---|---|
| suffix (linear) | 0.252 | 0.255 | 0.238 | 0.293 | 0.167 | 0.250 | 0.281 | 0.204 | 0.230 | 0.243 | 0.554 | 0.217 | 0.296 |
| suffix (tree) | 0.235 | 0.230 | 0.195 | 0.272 | 0.153 | 0.222 | 0.250 | 0.183 | 0.217 | 0.216 | 0.539 | 0.196 | 0.274 |
| pld | 0.191 | 0.210 | 0.184 | 0.215 | 0.128 | 0.208 | 0.231 | 0.174 | 0.199 | 0.175 | 0.379 | 0.175 | 0.225 |
| recycling | 0.028 | 0.027 | 0.026 | 0.028 | - | - | 0.025 | 0.027 | 0.025 | 0.026 | 0.026 | - | 0.026 |
| vanilla | - | - | - | - | - | - | - | - | - | - | - | - | - |
| suffix-1.0-tree | 0.385 | 0.381 | 0.348 | 0.420 | 0.330 | 0.388 | 0.409 | 0.349 | 0.375 | 0.375 | 0.632 | 0.373 | 0.421 |

## SWE-Bench: Time per output token (ms)

| System | astropy | django | matplotlib | seaborn | flask | requests | xarray | pylint | pytest | scikit-learn | sphinx | sympy | Overall |
|---|---|---|---|---|---|---|---|---|---|---|---|---|---|
| suffix (linear) | 30.563 | 21.881 | 26.013 | 37.012 | 22.238 | 23.828 | 23.598 | 27.739 | 29.313 | 31.949 | 25.613 | 17.072 | 27.436 |
| suffix (tree) | 30.800 | 22.238 | 26.278 | 37.095 | 22.728 | 23.998 | 23.847 | 28.388 | 29.451 | 32.496 | 25.853 | 17.199 | 27.711 |
| pld | 41.675 | 31.029 | 34.447 | 47.588 | 31.549 | 31.155 | 31.330 | 37.386 | 38.238 | 41.531 | 41.877 | 25.142 | 37.758 |
| recycling | 41.328 | 31.847 | 34.155 | 49.670 | - | - | 33.434 | 34.304 | 38.438 | 39.667 | 78.153 | - | 43.774 |
| vanilla | 50.080 | 41.102 | 43.019 | 57.069 | 35.040 | 39.268 | 41.213 | 42.107 | 45.633 | 46.061 | 77.815 | 32.527 | 50.074 |
| eagle3 | - | - | - | - | - | - | - | - | - | - | - | - | - |
| eagle2 | - | - | - | - | - | - | - | - | - | - | - | - | - |
| eagle | - | - | - | - | - | - | - | - | - | - | - | - | - |
| hybrid | - | - | - | - | - | - | - | - | - | - | - | - | - |

## SWE-Bench: Speculation time per generated token (ms)

| System | astropy | django | matplotlib | seaborn | flask | requests | xarray | pylint | pytest | scikit-learn | sphinx | sympy | Overall |
|---|---|---|---|---|---|---|---|---|---|---|---|---|---|
| vanilla | 0.000 | 0.000 | 0.000 | 0.000 | 0.000 | 0.000 | 0.000 | 0.000 | 0.000 | 0.000 | 0.000 | 0.000 | 0.000 |
| suffix (linear) | 0.156 | 0.122 | 0.175 | 0.185 | 0.198 | 0.182 | 0.178 | 0.203 | 0.217 | 0.203 | 0.125 | 0.167 | 0.171 |
| suffix (tree) | 0.170 | 0.135 | 0.172 | 0.172 | 0.191 | 0.165 | 0.165 | 0.235 | 0.235 | 0.212 | 0.138 | 0.150 | 0.175 |
| pld | 0.191 | 0.197 | 0.208 | 0.176 | 0.266 | 0.200 | 0.191 | 0.222 | 0.208 | 0.228 | 0.126 | 0.217 | 0.191 |
| recycling | 9.298 | 6.023 | 7.982 | 13.731 | - | - | 6.607 | 7.665 | 8.081 | 11.748 | 19.653 | - | 10.582 |
| eagle3 | - | - | - | - | - | - | - | - | - | - | - | - | - |
| eagle2 | - | - | - | - | - | - | - | - | - | - | - | - | - |
| eagle | - | - | - | - | - | - | - | - | - | - | - | - | - |
| hybrid | - | - | - | - | - | - | - | - | - | - | - | - | - |

## SWE-Bench: Speedup over vanilla decoding

| System | astropy | django | matplotlib | seaborn | flask | requests | xarray | pylint | pytest | scikit-learn | sphinx | sympy | Overall |
|---|---|---|---|---|---|---|---|---|---|---|---|---|---|
| suffix (linear) | 2.213 | 2.442 | 2.039 | 1.951 | 1.792 | 1.962 | 2.210 | 1.792 | 1.888 | 2.356 | 4.453 | 2.292 | 2.452 |
| suffix (tree) | 2.158 | 2.412 | 1.991 | 1.985 | 1.759 | 1.972 | 2.190 | 1.754 | 1.912 | 2.296 | 4.417 | 2.280 | 2.433 |
| pld | 1.440 | 1.516 | 1.425 | 1.360 | 1.242 | 1.399 | 1.486 | 1.270 | 1.326 | 1.429 | 2.006 | 1.483 | 1.495 |
| recycling | 1.427 | 1.458 | 1.483 | 1.264 | - | - | 1.360 | 1.311 | 1.290 | 1.399 | 1.328 | - | 1.358 |
| vanilla | 1.000 | 1.000 | 1.000 | 1.000 | 1.000 | 1.000 | 1.000 | 1.000 | 1.000 | 1.000 | 1.000 | 1.000 | 1.000 |
| eagle3 | - | - | - | - | - | - | - | - | - | - | - | - | - |
| eagle2 | - | - | - | - | - | - | - | - | - | - | - | - | - |
| eagle | - | - | - | - | - | - | - | - | - | - | - | - | - |
| hybrid | - | - | - | - | - | - | - | - | - | - | - | - | - |

**AgenticSQL: Mean accepted tokens (tokens/step)**

| System | Classify | Extract | Enrich | Combine | SQL1 | SQL2 | SQL3 | Overall |
|---|---|---|---|---|---|---|---|---|
| hybrid (tree) | 4.180 | 14.813 | 15.081 | 6.448 | 3.983 | 3.932 | 4.006 | 7.500 |
| hybrid (linear) | 4.008 | 13.473 | 15.304 | 6.362 | 3.876 | 3.842 | 3.913 | 7.262 |
| suffix (linear) | 3.577 | 11.833 | 12.395 | 5.924 | 3.665 | 3.005 | 4.005 | 6.349 |
| suffix (tree) | 3.470 | 11.724 | 12.137 | 5.834 | 3.633 | 2.914 | 3.904 | 6.236 |
| eagle2 | 3.156 | 5.173 | 3.249 | 3.534 | 3.066 | 3.761 | 3.057 | 3.572 |
| recycling | 2.915 | 4.125 | 3.125 | 3.138 | 2.951 | 2.929 | 2.994 | 3.169 |
| eagle3 | 2.529 | 3.374 | 4.198 | 3.179 | 2.142 | 4.622 | 2.056 | 3.160 |
| eagle | 2.305 | 4.062 | 2.877 | 3.127 | 2.166 | 3.295 | 2.109 | 2.851 |
| pld | 1.427 | 4.134 | 1.455 | 3.914 | 2.074 | 1.452 | 2.151 | 2.373 |
| vanilla | 1.000 | 1.000 | 1.000 | 1.000 | 1.000 | 1.000 | 1.000 | 1.000 |

**AgenticSQL: Mean Acceptance Rate**

| System | Classify | Extract | Enrich | Combine | SQL1 | SQL2 | SQL3 | Overall |
|---|---|---|---|---|---|---|---|---|
| vanilla | - | - | - | - | - | - | - | - |
| suffix (linear) | 0.212 | 0.628 | 0.740 | 0.428 | 0.318 | 0.245 | 0.330 | 0.415 |
| suffix (tree) | 0.189 | 0.605 | 0.614 | 0.397 | 0.294 | 0.225 | 0.298 | 0.375 |
| hybrid (tree) | 0.131 | 0.642 | 0.683 | 0.249 | 0.138 | 0.143 | 0.140 | 0.304 |
| hybrid (linear) | 0.124 | 0.595 | 0.750 | 0.242 | 0.130 | 0.139 | 0.133 | 0.302 |
| pld | 0.061 | 0.365 | 0.069 | 0.355 | 0.137 | 0.076 | 0.144 | 0.173 |
| eagle | 0.052 | 0.122 | 0.075 | 0.085 | 0.047 | 0.092 | 0.044 | 0.074 |
| eagle2 | 0.036 | 0.070 | 0.037 | 0.042 | 0.034 | 0.046 | 0.034 | 0.043 |
| eagle3 | 0.025 | 0.040 | 0.053 | 0.036 | 0.019 | 0.060 | 0.018 | 0.036 |
| recycling | 0.025 | 0.041 | 0.028 | 0.027 | 0.025 | 0.024 | 0.026 | 0.028 |

**AgenticSQL: Time per output token (ms)**

| System | Classify | Extract | Enrich | Combine | SQL1 | SQL2 | SQL3 | Overall |
|---|---|---|---|---|---|---|---|---|
| suffix (linear) | 9.552 | 2.687 | 3.007 | 6.023 | 9.559 | 10.588 | 9.767 | 7.306 |
| suffix (tree) | 9.594 | 2.876 | 3.188 | 6.164 | 10.339 | 10.935 | 10.090 | 7.592 |
| hybrid (tree) | 11.975 | 3.491 | 3.633 | 7.883 | 14.329 | 11.600 | 14.399 | 9.604 |
| recycling | 10.316 | 7.314 | 9.470 | 9.724 | 10.981 | 9.981 | 10.702 | 9.782 |
| hybrid (linear) | 12.563 | 3.681 | 3.762 | 8.603 | 14.827 | 11.639 | 21.219 | 10.874 |
| eagle2 | 16.814 | 9.210 | 15.078 | 13.877 | 17.967 | 12.312 | 17.531 | 14.677 |
| pld | 20.146 | 7.735 | 20.321 | 8.173 | 14.605 | 20.546 | 14.646 | 15.169 |
| eagle | 21.221 | 11.308 | 15.760 | 15.688 | 23.353 | 13.396 | 24.595 | 17.887 |
| eagle3 | 23.334 | 14.729 | 12.513 | 16.863 | 26.997 | 10.747 | 27.728 | 18.966 |
| vanilla | 26.578 | 25.292 | 25.032 | 25.406 | 27.011 | 25.851 | 26.307 | 25.924 |

**AgenticSQL: Speculation time per generated token (ms)**

| System | Classify | Extract | Enrich | Combine | SQL1 | SQL2 | SQL3 | Overall |
|---|---|---|---|---|---|---|---|---|
| vanilla | 0.000 | 0.000 | 0.000 | 0.000 | 0.000 | 0.000 | 0.000 | 0.000 |
| suffix (linear) | 0.060 | 0.015 | 0.015 | 0.033 | 0.059 | 0.058 | 0.061 | 0.043 |
| suffix (tree) | 0.064 | 0.017 | 0.020 | 0.035 | 0.064 | 0.062 | 0.064 | 0.047 |
| pld | 0.419 | 0.124 | 0.422 | 0.130 | 0.281 | 0.434 | 0.276 | 0.298 |
| recycling | 0.762 | 0.303 | 0.399 | 0.577 | 0.928 | 0.260 | 0.922 | 0.592 |
| hybrid (tree) | 1.728 | 0.249 | 0.376 | 1.112 | 2.350 | 0.927 | 2.372 | 1.299 |
| hybrid (linear) | 1.776 | 0.256 | 0.358 | 1.177 | 2.387 | 0.930 | 4.389 | 1.604 |
| eagle | 3.189 | 1.693 | 2.423 | 2.372 | 3.423 | 2.197 | 3.616 | 2.700 |
| eagle2 | 4.118 | 2.006 | 3.292 | 3.308 | 4.599 | 2.597 | 4.509 | 3.487 |
| eagle3 | 6.092 | 3.526 | 3.021 | 4.350 | 7.156 | 2.543 | 7.332 | 4.854 |

**AgenticSQL: Speedup over vanilla decoding**

| System | Classify | Extract | Enrich | Combine | SQL1 | SQL2 | SQL3 | Overall |
|---|---|---|---|---|---|---|---|---|
| suffix (linear) | 3.016 | 9.854 | 10.406 | 4.848 | 3.205 | 2.839 | 3.211 | 5.345 |
| suffix (tree) | 2.998 | 9.545 | 10.009 | 4.765 | 3.008 | 2.733 | 3.133 | 5.175 |
| hybrid (tree) | 2.338 | 7.672 | 8.191 | 3.613 | 2.057 | 2.496 | 2.077 | 4.068 |
| hybrid (linear) | 2.243 | 7.137 | 7.965 | 3.327 | 1.993 | 2.483 | 1.405 | 3.799 |
| recycling | 2.588 | 3.472 | 2.672 | 2.640 | 2.502 | 2.604 | 2.492 | 2.710 |
| pld | 1.330 | 3.695 | 1.255 | 3.298 | 1.936 | 1.311 | 1.905 | 2.105 |
| eagle2 | 1.591 | 2.751 | 1.673 | 1.855 | 1.527 | 2.119 | 1.527 | 1.864 |
| eagle3 | 1.328 | 1.720 | 2.025 | 1.619 | 1.108 | 2.496 | 1.056 | 1.623 |
| eagle | 1.324 | 2.246 | 1.598 | 1.669 | 1.229 | 1.955 | 1.138 | 1.595 |
| vanilla | 1.000 | 1.000 | 1.000 | 1.000 | 1.000 | 1.000 | 1.000 | 1.000 |

## Spec-Bench: Mean accepted tokens (tokens/step)

| System | coding | extraction | humanities | math | math_reasoning | qa | rag | reasoning | roleplay | stem | summarization | translation | writing | Overall |
|---|---|---|---|---|---|---|---|---|---|---|---|---|---|---|
| hybrid (tree) | 6.325 | 5.418 | 4.980 | 5.817 | 5.080 | 4.958 | 4.812 | 4.610 | 4.556 | 5.155 | 4.550 | 3.432 | 5.306 | 4.684 |
| eagle3 | 5.975 | 5.373 | 5.180 | 5.745 | 5.562 | 4.351 | 5.009 | 4.956 | 4.664 | 5.299 | 4.767 | 2.909 | 5.093 | 4.647 |
| hybrid (linear) | 5.958 | 5.249 | 4.966 | 5.446 | 5.121 | 4.452 | 4.753 | 4.607 | 4.557 | 5.089 | 4.520 | 3.345 | 5.158 | 4.553 |
| eagle2 | 4.766 | 3.842 | 3.612 | 4.242 | 4.129 | 3.324 | 3.630 | 3.892 | 3.353 | 3.736 | 3.254 | 2.605 | 3.383 | 3.466 |
| eagle | 4.149 | 3.469 | 3.177 | 3.724 | 3.617 | 2.857 | 3.239 | 3.328 | 2.968 | 3.311 | 2.926 | 2.352 | 3.082 | 3.065 |
| recycling | 3.044 | 2.610 | 2.539 | 3.128 | 2.980 | 2.372 | 2.352 | 2.537 | 2.338 | 2.697 | 2.614 | 2.305 | 2.417 | 2.548 |
| suffix (linear) | 1.981 | 1.757 | 1.454 | 2.161 | 1.661 | 1.878 | 1.999 | 1.521 | 1.252 | 1.485 | 1.725 | 1.705 | 1.435 | 1.766 |
| suffix (tree) | 1.960 | 1.754 | 1.461 | 2.134 | 1.638 | 1.874 | 1.957 | 1.504 | 1.259 | 1.501 | 1.703 | 1.705 | 1.424 | 1.750 |
| pld | 1.911 | 1.670 | 1.387 | 1.957 | 1.475 | 1.461 | 1.967 | 1.490 | 1.206 | 1.430 | 1.816 | 1.362 | 1.394 | 1.606 |
| vanilla | 1.000 | 1.000 | 1.000 | 1.000 | 1.000 | 1.000 | 1.000 | 1.000 | 1.000 | 1.000 | 1.000 | 1.000 | 1.000 | 1.000 |

## Spec-Bench: Mean Acceptance Rate

| System | coding | extraction | humanities | math | math_reasoning | qa | rag | reasoning | roleplay | stem | summarization | translation | writing | Overall |
|---|---|---|---|---|---|---|---|---|---|---|---|---|---|---|
| suffix (linear) | 0.255 | 0.216 | 0.162 | 0.240 | 0.163 | 0.152 | 0.216 | 0.171 | 0.116 | 0.169 | 0.199 | 0.216 | 0.208 | 0.190 |
| vanilla | - | - | - | - | - | - | - | - | - | - | - | - | - | - |
| suffix (tree) | 0.244 | 0.211 | 0.156 | 0.232 | 0.149 | 0.144 | 0.203 | 0.156 | 0.115 | 0.165 | 0.189 | 0.208 | 0.194 | 0.179 |
| pld | 0.132 | 0.097 | 0.063 | 0.126 | 0.071 | 0.088 | 0.119 | 0.076 | 0.037 | 0.068 | 0.103 | 0.130 | 0.072 | 0.099 |
| eagle | 0.126 | 0.099 | 0.087 | 0.109 | 0.105 | 0.074 | 0.090 | 0.093 | 0.079 | 0.092 | 0.077 | 0.054 | 0.083 | 0.083 |
| hybrid (linear) | 0.109 | 0.084 | 0.072 | 0.100 | 0.078 | 0.074 | 0.075 | 0.069 | 0.062 | 0.075 | 0.068 | 0.044 | 0.077 | 0.070 |
| hybrid (tree) | 0.107 | 0.084 | 0.072 | 0.101 | 0.077 | 0.075 | 0.074 | 0.069 | 0.062 | 0.076 | 0.068 | 0.045 | 0.077 | 0.070 |
| eagle3 | 0.083 | 0.073 | 0.070 | 0.079 | 0.076 | 0.056 | 0.067 | 0.066 | 0.061 | 0.072 | 0.063 | 0.032 | 0.068 | 0.061 |
| eagle2 | 0.063 | 0.047 | 0.044 | 0.054 | 0.052 | 0.039 | 0.044 | 0.048 | 0.039 | 0.046 | 0.038 | 0.027 | 0.040 | 0.041 |
| recycling | 0.026 | 0.020 | 0.019 | 0.027 | 0.025 | 0.017 | 0.017 | 0.020 | 0.017 | 0.021 | 0.020 | 0.017 | 0.018 | 0.020 |

## Spec-Bench: Time per output token (ms)

| System | coding | extraction | humanities | math | math_reasoning | qa | rag | reasoning | roleplay | stem | summarization | translation | writing | Overall |
|---|---|---|---|---|---|---|---|---|---|---|---|---|---|---|
| hybrid (linear) | 7.218 | 9.235 | 8.970 | 7.974 | 8.783 | 11.534 | 10.576 | 9.899 | 10.195 | 8.698 | 9.857 | 14.858 | 8.806 | 10.747 |
| hybrid (tree) | 7.251 | 9.588 | 9.519 | 8.203 | 9.194 | 11.880 | 11.098 | 10.543 | 10.775 | 9.133 | 10.214 | 15.262 | 9.141 | 11.153 |
| recycling | 9.475 | 11.528 | 11.333 | 9.224 | 9.749 | 13.475 | 12.957 | 11.489 | 12.425 | 10.691 | 11.373 | 13.104 | 12.019 | 11.947 |
| eagle2 | 9.721 | 13.256 | 12.822 | 11.157 | 11.338 | 14.903 | 14.012 | 12.631 | 14.229 | 12.461 | 14.422 | 19.116 | 13.992 | 14.387 |
| eagle | 10.201 | 13.193 | 13.570 | 11.485 | 11.895 | 16.219 | 14.222 | 13.215 | 14.639 | 13.022 | 14.840 | 19.425 | 14.280 | 14.925 |
| suffix (tree) | 13.833 | 16.284 | 18.670 | 13.547 | 16.304 | 19.839 | 15.285 | 18.382 | 21.339 | 18.034 | 15.825 | 16.595 | 19.154 | 16.875 |
| suffix (linear) | 13.595 | 16.245 | 18.212 | 13.292 | 16.447 | 19.999 | 15.378 | 18.061 | 21.217 | 18.041 | 15.925 | 16.670 | 18.925 | 16.936 |
| pld | 14.681 | 17.524 | 20.469 | 14.450 | 19.179 | 22.693 | 16.383 | 19.173 | 23.300 | 19.714 | 15.949 | 22.214 | 20.438 | 19.190 |
| vanilla | 24.721 | 24.903 | 24.979 | 24.466 | 24.992 | 25.082 | 25.698 | 24.475 | 24.433 | 24.871 | 25.114 | 24.904 | 24.463 | 25.076 |

## Spec-Bench: Speculation time per generated token (ms)

| System | coding | extraction | humanities | math | math_reasoning | qa | rag | reasoning | roleplay | stem | summarization | translation | writing | Overall |
|---|---|---|---|---|---|---|---|---|---|---|---|---|---|---|
| vanilla | 0.000 | 0.000 | 0.000 | 0.000 | 0.000 | 0.000 | 0.000 | 0.000 | 0.000 | 0.000 | 0.000 | 0.000 | 0.000 | 0.000 |
| suffix (linear) | 0.071 | 0.079 | 0.099 | 0.067 | 0.087 | 0.097 | 0.089 | 0.084 | 0.101 | 0.094 | 0.092 | 0.076 | 0.084 | 0.088 |
| suffix (tree) | 0.074 | 0.080 | 0.104 | 0.069 | 0.091 | 0.099 | 0.093 | 0.088 | 0.103 | 0.097 | 0.096 | 0.079 | 0.086 | 0.091 |
| recycling | 0.243 | 0.318 | 0.291 | 0.240 | 0.255 | 0.391 | 0.419 | 0.304 | 0.320 | 0.275 | 0.321 | 0.365 | 0.310 | 0.340 |
| pld | 0.291 | 0.357 | 0.421 | 0.277 | 0.392 | 0.480 | 0.326 | 0.391 | 0.492 | 0.407 | 0.325 | 0.466 | 0.428 | 0.395 |
| hybrid (linear) | 1.369 | 1.838 | 2.026 | 1.510 | 1.854 | 2.455 | 2.152 | 2.070 | 2.353 | 1.925 | 2.118 | 3.096 | 1.949 | 2.259 |
| eagle | 1.698 | 1.977 | 2.268 | 1.877 | 1.892 | 2.389 | 2.034 | 2.060 | 2.421 | 2.166 | 2.434 | 2.885 | 2.351 | 2.289 |
| hybrid (tree) | 1.445 | 1.935 | 2.152 | 1.580 | 1.947 | 2.553 | 2.238 | 2.506 | 2.010 | 2.200 | 3.141 | 2.056 | 2.344 |
| eagle3 | 1.913 | 2.172 | 2.218 | 2.001 | 2.030 | 2.674 | 2.419 | 2.334 | 2.500 | 2.160 | 2.402 | 4.084 | 2.237 | 2.634 |
| eagle2 | 2.055 | 2.656 | 2.714 | 2.338 | 2.346 | 2.962 | 2.807 | 2.584 | 2.997 | 2.628 | 3.062 | 3.826 | 2.946 | 2.936 |

## Spec-Bench: Speedup over vanilla decoding

| System | coding | extraction | humanities | math | math_reasoning | qa | rag | reasoning | roleplay | stem | summarization | translation | writing | Overall |
|---|---|---|---|---|---|---|---|---|---|---|---|---|---|---|
| hybrid (linear) | 3.441 | 2.795 | 2.800 | 3.139 | 2.866 | 2.404 | 2.542 | 2.496 | 2.461 | 2.871 | 2.573 | 1.759 | 2.833 | 2.500 |
| hybrid (tree) | 3.442 | 2.717 | 2.644 | 3.132 | 2.736 | 2.611 | 2.445 | 2.344 | 2.330 | 2.732 | 2.485 | 1.713 | 2.736 | 2.458 |
| eagle3 | 3.115 | 2.634 | 2.713 | 2.910 | 2.888 | 2.172 | 2.474 | 2.446 | 2.394 | 2.764 | 2.520 | 1.447 | 2.622 | 2.367 |
| recycling | 2.619 | 2.234 | 2.209 | 2.660 | 2.577 | 1.951 | 2.057 | 2.150 | 1.979 | 2.334 | 2.218 | 1.932 | 2.057 | 2.169 |
| eagle2 | 2.548 | 1.998 | 1.958 | 2.215 | 2.220 | 1.733 | 1.877 | 1.968 | 1.751 | 2.007 | 1.756 | 1.336 | 1.791 | 1.825 |
| eagle | 2.427 | 1.956 | 1.848 | 2.142 | 2.112 | 1.600 | 1.841 | 1.871 | 1.703 | 1.922 | 1.702 | 1.305 | 1.753 | 1.752 |
| suffix (tree) | 1.796 | 1.620 | 1.350 | 1.930 | 1.558 | 1.785 | 1.886 | 1.364 | 1.149 | 1.389 | 1.624 | 1.627 | 1.310 | 1.661 |
| suffix (linear) | 1.828 | 1.630 | 1.383 | 1.967 | 1.544 | 1.773 | 1.890 | 1.389 | 1.156 | 1.386 | 1.618 | 1.618 | 1.327 | 1.659 |
| pld | 1.712 | 1.484 | 1.231 | 1.753 | 1.323 | 1.326 | 1.807 | 1.320 | 1.055 | 1.275 | 1.628 | 1.219 | 1.232 | 1.448 |
| vanilla | 1.000 | 1.000 | 1.000 | 1.000 | 1.000 | 1.000 | 1.000 | 1.000 | 1.000 | 1.000 | 1.000 | 1.000 | 1.000 | 1.000 |

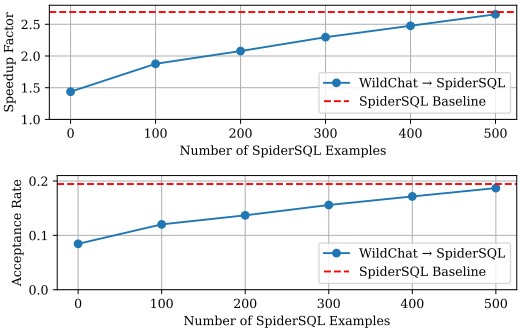

Figure 9: The performance of SuffixDecoding under input distribution shift. SuffixDecoding was trained on outputs from WildChat, while being evaluated on SpiderSQL. X axis: the number of SpiderSQL inputs, which are added to the global suffix tree after they are processed. Red line: the performance of SuffixDecoding if trained on 500 output examples from only SpiderSQL offline.

## A.2   Additional Ablation Experiments

In this appendix, we share ablation studies that reveal the impact of several design decisions in SuffixDecoding. The studies are conducted using the simulated verifier described in Sec. 4.1.

### A.2.1   Additional Dataset Details

We performed additional ablation experiments, which used additional datasets described below.

1. *WildChat* [Zhao et al., 2024]. We use instructions from the WildChat dataset, which consists of real-world interactions between users and the ChatGPT service. WildChat represents the most diverse and open-domain dataset used in our evaluations.

2. *Magicoder* [Wei et al., 2023]. Specifically, we use instructions from the Magicoder-Evol-Instruct-110K dataset, which consists of code-related questions and instructions generated via Self-Instruct [Chaudhary, 2023, Wang et al., 2023b] and further augmented for difficulty and diversity [Luo et al., 2023] by GPT-4.

3. *SpiderSQL*. Spider [Yu et al., 2018] is a dataset of manually-annotated questions and SQL responses over 200 different databases with multiple tables, covering 138 different domains. We use instructions from DAIL-SQL [Gao et al., 2024a], which consists of LLM prompts with instructions to answer questions from Spider using structured SQL code.

### A.2.2   Effect of input distribution shift

In real-world LLM serving, the input characteristics of requests may change over time, and may be out-of-distribution from the output examples that SuffixDecoding was trained on. To evaluate this scenario, we run SuffixDecoding trained on WildChat outputs, and begin to send it inputs from SpiderSQL, which represents a very sudden distribution shift.

Fig. 9 shows the results. SuffixDecoding starts from having 4,000 output examples from WildChat, and begins to receive SpiderSQL inference requests. Without any adaptation, SuffixDecoding still achieves $1.5\times$ speedup and $8\%$ acceptance rate, but is far from the $2.6\times$ speedup and $20\%$ acceptance rate it would achieve if it were trained on 500 examples from SpiderSQL instead.

After processing each SpiderSQL inference request, SuffixDecoding can insert its output into its global suffix tree, which means it can adapt in an online fashion to the new input distribution. As Fig. 9 shows, the performance of SuffixDecoding improves with the number of SpiderSQL inference requests processed. Perhaps surprisingly, after observing 500 SpiderSQL and adapting online, SuffixDecoding's performance is almost indistinguishable to its performance if it were trained offline on the 500 SpiderSQL examples alone. This suggests that SuffixDecoding is able to adapt to input distribution shifts quickly and at no loss in performance.

### A.2.3 Predicting SuffixDecoding Effectiveness

SuffixDecoding tends to perform better on more structured tasks compared to very open-ended ones (e.g., AgenticSQL vs WildChat). We can measure this "structuredness" using empirical entropy. The steps are as follows: (1) create a suffix tree from example model outputs (100 examples is typically enough), (2) calculate the entropy of each node's output distribution by determining how often each child node is accessed, and (3) compute a weighted average of this entropy across all nodes. A low average entropy indicates that output tokens are more predictable based on their prefixes, which generally suggests that Suffix Decoding will perform better.

Table 1: Measured empirical entropy of our various evaluation datasets.

| Dataset | Average Entropy |
|---|---|
| AgenticSQL (Enrich) | 0.171 |
| AgenticSQL (Classify) | 0.738 |
| AgenticSQL (Extract) | 0.0862 |
| AgenticSQL (SQL1) | 1.52 |
| AgenticSQL (SQL2) | 1.49 |
| AgenticSQL (SQL3) | 1.51 |
| AgenticSQL (Combine) | 1.49 |
| Spider | 2.50 |
| WildChat | 3.43 |
| Magicoder | 2.95 |

Table 1 shows the empirical entropy measured on samples from each of our evaluation datasets. We find that the average entropy is closely related to the intuitive understanding of the "structuredness" of each dataset. Additionally, it correlates well with the performance of SuffixDecoding on those datasets. Therefore, practitioners can calculate this value using a small number of output examples to assess whether SuffixDecoding is appropriate for their specific tasks.

## B  Scalability and Memory Overhead

The suffix trees employed by Suffix Decoding are highly time and memory efficient. Table 2 shows the per-token lookup time, update time, and memory consumption across various tree sizes (using requests from the SWE-Bench dataset).

Table 2: Performance metrics for different entry counts

| Total # entries | Total # tokens | Memory footprint (MB) | Avg update time per token ($\mu s$) | Avg lookup time per speculated token ($\mu s$) |
|---|---|---|---|---|
| 1 K entries | 27 M | 1670 MB | 3.95 | 12.18 |
| 5 K entries | 143 M | 2980 MB | 4.06 | 10.93 |
| 10 K entries | 285 M | 4070 MB | 4.05 | 12.00 |
| 15 K entries | 429 M | 5110 MB | 4.07 | 11.62 |
| 20 K entries | 572 M | 6150 MB | 4.04 | 11.64 |

Overall, the total memory consumption scales linearly with the size of the tree, and both the update and lookup times remain fast even with larger trees. Given typical server configurations, SuffixDecoding can cache many weeks of generated tokens before hitting CPU memory limits. For example, optimized inference engines running Llama-8B on an A100 GPU can typically generate $\sim 5000$ tokens per second ([Yin et al., 2024]) or $\sim 432M$ tokens per day. Typical A100 systems (e.g. AWS p4d.24xlarge) have 144GB of CPU memory per A100 GPU. This means that Suffix Decoding can potentially cache 31 days (see math below) of generated tokens per A100 GPU before hitting CPU memory limits. Beyond this limit, it is also straightforward to incorporate cache eviction (e.g. LRU) into Suffix Decoding to avoid out-of-memory problems.

**Time to fill up GPU memory:**

1. Memory per token: $6.15$ GB $\div$ 572M tokens $= 1.075 \times 10^{-8}$ GB/token

2. Total token capacity: $144 \text{ GB} \div (1.075 \times 10^{-8} \text{ GB/token}) = 1.34 \times 10^{10}$ tokens

3. Time to fill: $(1.34 \times 10^{10} \text{ tokens}) \div (5000 \text{ tok/s}) = 2.68 \times 10^{6}$ seconds = **31.0** days

## B.1 Hybrid Fallback Threshold Sensitivity Analysis

The hybrid SuffixDecoding approach uses a fallback threshold $\tau$ to determine when to use suffix-tree-based speculation versus falling back to a model-based speculator (e.g., EAGLE-3). This section analyzes the sensitivity of performance to different threshold values across both open-ended and agentic workloads.

Table 3 shows the wall-clock speedup results on Spec-Bench for various threshold values. For open-ended generation tasks, we find that setting $\tau$ close to or slightly exceeding the mean accepted tokens (MAT) of the model-based speculator yields the best performance. For instance, EAGLE-3 achieves a MAT of approximately 4.65 tokens/step on Spec-Bench. The results show that $\tau \in [5, 7]$ produces the highest overall speedups (2.5×), with performance being relatively stable across this range.

Table 3: Hybrid SuffixDecoding speedup on Spec-Bench across different threshold values $\tau$.

| $\tau$ | coding | qa | math | reasoning | stem | writing | Overall |
|---|---|---|---|---|---|---|---|
| 7 | 3.44 | 2.40 | 3.14 | 2.50 | 2.87 | 2.83 | 2.50 |
| 6 | 3.19 | 2.22 | 2.88 | 2.30 | 2.68 | 2.61 | 2.31 |
| 5 | 3.25 | 2.28 | 2.70 | 2.35 | 2.74 | 2.70 | 2.37 |
| 4 | 3.07 | 2.22 | 2.72 | 2.25 | 2.57 | 2.64 | 2.25 |
| 3 | 2.88 | 2.09 | 2.54 | 2.10 | 2.39 | 2.50 | 2.13 |
| 2 | 2.73 | 2.04 | 2.40 | 2.01 | 2.19 | 2.36 | 2.03 |
| 1 | 2.28 | 1.86 | 2.01 | 1.69 | 1.90 | 2.12 | 1.80 |
| 0 (suffix only) | 1.83 | 1.77 | 1.97 | 1.39 | 1.39 | 1.33 | 1.66 |

Conversely, Table 4 shows results for AgenticSQL, a highly repetitive agentic workload. Here, the standalone SuffixDecoding ($\tau = 0$) achieves the best overall speedup (5.35×), as it can confidently speculate much longer sequences than model-based methods. Higher threshold values progressively degrade performance as the system increasingly falls back to less effective model-based speculation.

Table 4: Hybrid SuffixDecoding speedup on AgenticSQL across different threshold values $\tau$.

| $\tau$ | Classify | Extract | Enrich | Combine | SQL | Overall |
|---|---|---|---|---|---|---|
| 0 (suffix only) | 3.02 | 9.85 | 10.41 | 4.85 | 3.09 | 5.35 |
| 1 | 2.31 | 7.03 | 8.37 | 3.38 | 2.17 | 3.95 |
| 2 | 2.24 | 7.14 | 7.97 | 3.33 | 2.29 | 3.80 |
| 3 | 2.22 | 5.51 | 5.68 | 3.52 | 2.20 | 3.22 |
| 7 | 2.06 | 6.79 | 7.10 | 3.44 | 2.29 | 3.66 |

In summary, practitioners should set $\tau \approx \text{MAT}_{\text{fallback}}$ for mixed or open-ended workloads, where $\text{MAT}_{\text{fallback}}$ is the mean accepted tokens of the fallback model-based speculator. For highly repetitive agentic workloads, using SuffixDecoding alone ($\tau = 0$ or very low values) yields the best performance.

## C Batch-level Speculation Control

For batched serving scenarios, optimizing the speculation per request in a batch is crucial for many practical online deployments. While SuffixDecoding focuses on *what* tokens to speculate, it is also compatible with existing works that explore *how much* to speculate per request. For example, TurboSpec [Liu et al., 2024] uses a "goodput" metric to balance speculation lengths and batch size, noting that optimal speculation decreases as batch size increases. AdaServe [Li et al., 2025b] formulates multi-SLO serving as a constrained optimization problem and introduces SLO-customized speculative decoding that constructs speculation trees tailored to each request's latency target. By dynamically adjusting the number of speculative tokens generated by the SuffixTree per request,

SuffixDecoding can be integrated with TurboSpec to maximize the batch-wise goodput metric. Furthermore, the statistics-based scoring used by Suffix Decoding can potentially better inform methods like TurboSpec. For example, choosing to speculate more from sequences that have a higher marginal score. These are interesting and important directions for future work.

