# OpenReview forum: "SuffixDecoding: Extreme Speculative Decoding for Emerging AI Applications"
_NeurIPS.cc/2025/Conference — NeurIPS 2025 spotlight_

### Official Review · Reviewer_Bnmw · 2025-06-29

**Clarity:** 4
**Significance:** 3
**Originality:** 3
**Rating:** 5
**Confidence:** 4

**Summary:**

This paper proposes a novel model-free speculative decoding algorithm for the LLM agent domain. By leveraging a prefix-tree, the proposed algorithm could effectively find previous tokens for the new speculation. Even though this method cannot be genealize to all domains, it serves well for the agent domain, where the generation often follows certain patterns. Overall, I think this paper proposes an efficient solution for this niche challenge.

**Questions:**

1. How will the proposed algorithm impact the memory consumption
2. What is the performance of the proposed algorithm on a more general domain benchmark like MT-Bench?

**Ethical Concerns:**

["NO or VERY MINOR ethics concerns only"]

**Final Justification:**

I have read the author's feedback and tend to keep my original rating of accepting this paper.

**Limitations:**

yes

**Quality:**

3

**Strengths And Weaknesses:**

Strengths:
1. The proposed method is sound and carefully designed for the target scenario
2. The paper is clearly written and easy to follow

Weakness:
1. The technical depth of this work can be further improved. For example, the current adaptive speculation length algorithm is quite straightforward. As the main contribution of the proposed algorithm, it is worth further investigation. For example, whether the past information in each trajectory can help select a better speculation length.

---

> ### Author Rebuttal · Authors · 2025-07-31
>
> Thank you for your positive review and detailed feedback, our response is below.
>
> **How will the proposed algorithm impact memory consumption?**
>
> Suffix Decoding uses CPU memory on the host during serving, which is usually plentiful and under-utilized. To show Suffix Decoding’s memory consumption and other performance stats, we performed an additional micro-benchmark on its suffix tree, shown below:
>
> | Total \# entries | Total \# tokens | Memory footprint (MB) | Avg update time per token (μs) | Avg lookup time per speculated token (μs) |
> | :---- | :---- | :---- | :---- | :---- |
> | 1 K entries | 27 M | 1670 MB | 3.95 | 12.18 |
> | 5 K entries | 143 M | 2980 MB | 4.06 | 10.93 |
> | 10 K entries | 285 M | 4070 MB | 4.05 | 12.00 |
> | 15 K entries | 429 M | 5110 MB | 4.07 | 11.62 |
> | 20 K entries | 572 M | 6150 MB | 4.04 | 11.64 |
>
> As shown, Suffix Decoding can cache 572M tokens using only 6.15 GB of CPU memory, and its memory usage scales linearly. In comparison, optimized serving systems can generate on the order of \~5000 tokens/s for Llama-8B on an A100 GPU \[2\]. Typical A100 systems (e.g. AWS p4d.24xlarge) have 144GB of CPU memory per A100 GPU. This means that Suffix Decoding can potentially cache 31 days of generated tokens (see math below) before requiring cache eviction.
>
> Time to fill up GPU memory:
>
> 1. Memory per token: $$\frac{6.15 GB}{572M tokens} = 1.075 \times 10^{-8} GB/token$$
> 2. Total token capacity: $$\frac{144 GB}{(1.075 \times 10^{-8} GB/token)} = 1.34 \times 10^{10} tokens$$
> 3. Time to fill RAM: $$\frac{1.34 \times 10^{10} tokens}{5000 tok/s} = 2.68 \times 10^{6} seconds =  \boxed{31 days}$$
>
> **General domain (e.g. MT-Bench) performance.**
>
> Suffix Decoding's MT-Bench performance is reflected in SpecBench A.1.2 tables. These tables contain 8 MT-Bench subsets (Writing, Roleplay, Reasoning, Math, Coding, Extraction, STEM, Humanities). As SpecBench equally samples all 8 categories, overall Suffix Decoding MT-Bench performance can be calculated by averaging these column results.
>
> Additionally, we provide evaluation results that include hybrid Suffix/Spec Decoding (end of Sec 3\) using Suffix Decoding with EAGLE-3:
>
> **SpecBench \- Speedup (x)**
>
> | System | coding | extraction | humanities | math | math\_reasoning | qa | rag | reasoning | roleplay | stem | summarization | translation | writing | Overall |
> | :---- | :---- | :---- | :---- | :---- | :---- | :---- | :---- | :---- | :---- | :---- | :---- | :---- | :---- | :---- |
> | hybrid (τ=7) | 3.441 | 2.795 | 2.800 | 3.139 | 2.866 | 2.404 | 2.542 | 2.496 | 2.461 | 2.871 | 2.573 | 1.759 | 2.833 | 2.500 |
> | eagle3 | 3.115 | 2.634 | 2.713 | 2.910 | 2.888 | 2.172 | 2.474 | 2.446 | 2.394 | 2.764 | 2.520 | 1.447 | 2.622 | 2.367 |
> | recycling | 2.619 | 2.234 | 2.209 | 2.660 | 2.577 | 1.951 | 2.057 | 2.150 | 1.979 | 2.334 | 2.218 | 1.932 | 2.057 | 2.169 |
> | eagle2 | 2.548 | 1.998 | 1.958 | 2.215 | 2.220 | 1.733 | 1.877 | 1.968 | 1.751 | 2.007 | 1.756 | 1.336 | 1.791 | 1.825 |
> | eagle | 2.427 | 1.956 | 1.848 | 2.142 | 2.112 | 1.600 | 1.841 | 1.871 | 1.703 | 1.922 | 1.702 | 1.305 | 1.753 | 1.752 |
> | suffix | 1.828 | 1.630 | 1.383 | 1.967 | 1.544 | 1.773 | 1.890 | 1.389 | 1.156 | 1.386 | 1.618 | 1.618 | 1.327 | 1.659 |
> | pld | 1.712 | 1.484 | 1.231 | 1.753 | 1.323 | 1.326 | 1.807 | 1.320 | 1.055 | 1.275 | 1.628 | 1.219 | 1.232 | 1.448 |
> | vanilla | 1.000 | 1.000 | 1.000 | 1.000 | 1.000 | 1.000 | 1.000 | 1.000 | 1.000 | 1.000 | 1.000 | 1.000 | 1.000 | 1.000 |
>
> **AgenticSQL \- Speedup (x)**
>
> | System | CATEGORIZATION | FEATURE\_EXTRACTION | QUESTION\_SUGGESTION | SQL\_COMBINE | SQL\_FANOUT1 | SQL\_FANOUT2 | SQL\_FANOUT3 | Overall |
> | :---- | :---- | :---- | :---- | :---- | :---- | :---- | :---- | :---- |
> | suffix | 3.016 | 9.854 | 10.406 | 4.848 | 3.205 | 2.839 | 3.211 | 5.345 |
> | hybrid (τ=1) | 2.309 | 7.025 | 8.368 | 3.382 | 2.024 | 2.421 | 2.072 | 3.947 |
> | recycling | 2.588 | 3.472 | 2.672 | 2.640 | 2.502 | 2.604 | 2.492 | 2.710 |
> | pld | 1.330 | 3.695 | 1.255 | 3.298 | 1.936 | 1.311 | 1.905 | 2.105 |
> | eagle2 | 1.591 | 2.751 | 1.673 | 1.855 | 1.527 | 2.119 | 1.527 | 1.864 |
> | eagle3 | 1.328 | 1.720 | 2.025 | 1.619 | 1.108 | 2.496 | 1.056 | 1.623 |
> | eagle | 1.324 | 2.246 | 1.598 | 1.669 | 1.229 | 1.955 | 1.138 | 1.595 |
> | vanilla | 1.000 | 1.000 | 1.000 | 1.000 | 1.000 | 1.000 | 1.000 | 1.000 |
>
> As shown, the hybrid Suffix Decoding and EAGLE-3 method performs similarly to plain Suffix Decoding on the more structured/repetitive tasks (e.g. AgenticSQL), while being faster than plain EAGLE-3 on the more general domain SpecBench. We will include these additional results in updating our paper.
>
> \[1\] TurboSpec: Closed-loop Speculation Control System for Optimizing LLM Serving Goodput | Liu et al. 2025 | arXiv:2406.14066
> \[2\] (Achieving Faster Open-Source Llama3 Serving with SGLang Runtime (vs. TensorRT-LLM, vLLM) | SGLang Team, Jul 25, 2024 | LMSYS Org)

---

### Official Review · Reviewer_QNpe · 2025-07-01

**Clarity:** 3
**Significance:** 3
**Originality:** 3
**Rating:** 5
**Confidence:** 3

**Summary:**

This paper introduces SuffixDecoding, a novel, model-free speculative decoding method designed to reduce LLM inference latency for agentic AI applications that produce repetitive and predictable sequences. SuffixDecoding uses efficient suffix trees to cache and rapidly match long token sequences from prompts and past outputs, enabling fast draft token generation without GPU overhead. A key innovation is its adaptive nature; it speculates more tokens when pattern matches are long and confidence is high, and fewer when matches are short, thereby conserving computational resources. Evaluations on agentic benchmarks like SWE-Bench and a Text-to-SQL pipeline show that SuffixDecoding achieves speedups of up to 3.9x, outperforming SOTA model-based (EAGLE-2/3) and model-free (Token Recycling) methods on these agentic tasks.

**Questions:**

It would be great if the authors could also briefly discuss how to trade off the number of draft tokens per request within a batch, especially when all requests might have high confidence in generating long draft sequences -- how the proposed method can be properly integrated with the online serving scenario.

**Ethical Concerns:**

["NO or VERY MINOR ethics concerns only"]

**Final Justification:**

The paper addresses an important problem and demonstrates superior performance. The author further discussed the possibility of integrating the method with the common serving scenario and provide results for open-ended tasks.

**Limitations:**

yes

**Quality:**

3

**Strengths And Weaknesses:**

Strengths:
1. The paper addresses the critical and timely issue of LLM inference latency, focusing on the important and growing domain of agentic applications.
2. The method proposed in the paper is simple and straightforward while being practically useful.
3. Significant speedup on agentic workloads compared to SOTA model-based and model-free methods.
4. The ablation experiments answer most of my questions about the method.

Weaknesses
1. The primary benefit of the proposed method is in repetitive workloads. While highly effective for agentic applications characterized by repetitive token sequences, its advantages may be less pronounced in purely open-ended or less repetitive scenarios when compared to specialized model-based methods designed for those tasks. However, the paper does show it can learn useful patterns even in open-ended workloads and can fall back to the SOTA model-based method.

---

> ### Author Rebuttal · Authors · 2025-07-31
>
> Thank you for your positive review and detailed feedback, our response is below.
>
> **Performance on open-ended or less repetitive token sequences.**
>
> As the reviewer mentioned, Suffix Decoding can still learn useful patterns even in open-ended workloads and can fall back to the SOTA model-based method. The hybrid suffix/speculative decoding (end of Sec 3\) also helps address this scenario. We provide additional evaluations that show the performance of this hybrid method below:
>
> **SpecBench \- Speedup (x)**
>
> | System | coding | extraction | humanities | math | math\_reasoning | qa | rag | reasoning | roleplay | stem | summarization | translation | writing | Overall |
> | :---- | :---- | :---- | :---- | :---- | :---- | :---- | :---- | :---- | :---- | :---- | :---- | :---- | :---- | :---- |
> | hybrid (τ=7) | 3.441 | 2.795 | 2.800 | 3.139 | 2.866 | 2.404 | 2.542 | 2.496 | 2.461 | 2.871 | 2.573 | 1.759 | 2.833 | 2.500 |
> | eagle3 | 3.115 | 2.634 | 2.713 | 2.910 | 2.888 | 2.172 | 2.474 | 2.446 | 2.394 | 2.764 | 2.520 | 1.447 | 2.622 | 2.367 |
> | recycling | 2.619 | 2.234 | 2.209 | 2.660 | 2.577 | 1.951 | 2.057 | 2.150 | 1.979 | 2.334 | 2.218 | 1.932 | 2.057 | 2.169 |
> | eagle2 | 2.548 | 1.998 | 1.958 | 2.215 | 2.220 | 1.733 | 1.877 | 1.968 | 1.751 | 2.007 | 1.756 | 1.336 | 1.791 | 1.825 |
> | eagle | 2.427 | 1.956 | 1.848 | 2.142 | 2.112 | 1.600 | 1.841 | 1.871 | 1.703 | 1.922 | 1.702 | 1.305 | 1.753 | 1.752 |
> | suffix | 1.828 | 1.630 | 1.383 | 1.967 | 1.544 | 1.773 | 1.890 | 1.389 | 1.156 | 1.386 | 1.618 | 1.618 | 1.327 | 1.659 |
> | pld | 1.712 | 1.484 | 1.231 | 1.753 | 1.323 | 1.326 | 1.807 | 1.320 | 1.055 | 1.275 | 1.628 | 1.219 | 1.232 | 1.448 |
> | vanilla | 1.000 | 1.000 | 1.000 | 1.000 | 1.000 | 1.000 | 1.000 | 1.000 | 1.000 | 1.000 | 1.000 | 1.000 | 1.000 | 1.000 |
>
> As shown, hybrid Suffix Decoding \+ EAGLE-3 performs similarly to plain Suffix Decoding on more structured/repetitive tasks (e.g. AgenticSQL), while slightly improving upon plain EAGLE-3 on the open-ended tasks (Spec-Bench). We will include these additional results in the updated version of our paper. Given space limitations, we are including here only the wall-clock speedups compared to the baseline. A comprehensive breakdown of results, including per-token latency (TPOT), mean accepted tokens per step (MAT), mean acceptance rate, and speculation time per generated token, will be incorporated into the revised paper.
>
> **How to set the number of draft tokens per request within a batch.**
>
> We agree that optimizing the speculation per request in a batch is crucial for many practical online deployments. While SuffixDecoding focuses on *what* tokens to speculate, it is also compatible with existing works that explore *how much* to speculate per request. For example, **TurboSpec** \[1\] uses a "goodput" metric to balance speculation lengths and batch size, noting that optimal speculation decreases as batch size increases. By dynamically adjusting the number of speculative tokens generated by the SuffixTree per request, SuffixDecoding can be integrated with TurboSpec to maximize the batch-wise goodput metric.
>
> Furthermore, the statistics-based scoring used by Suffix Decoding can potentially better inform methods like TurboSpec. For example, choosing to speculate more from sequences that have a higher marginal score. These are interesting and important directions for future work.
>
> [1] TurboSpec: Closed-loop Speculation Control System for Optimizing LLM Serving Goodput | Liu et al. 2025 | arXiv:2406.14066

---

### Official Review · Reviewer_CbVA · 2025-07-03

**Clarity:** 4
**Significance:** 4
**Originality:** 3
**Rating:** 5
**Confidence:** 3

**Summary:**

The paper introduces SuffixDecoding, a model-free speculative decoding method to save compute at inference time for LLMs by reusing tokens predicted on previous prompts/contexts.  The novelty introduced is in the trie/suffix tree which allows for lookup of past sequences (from  prompts or completions) in a dynamic / adaptive system.  Its highlighted use in agent-based systems with structured outputs is highlighted due to the type of repetition present in those settings that allows for the differentiated improvement of this method in storing/retrieving past tokens in such a system.  Like standard speculative decoding, the inference result is exact, with no accuracy loss, and only gains from leveraging the previous cache.  The experiments show significant speedups compared to existing methods in agent/structured settings.  The method has also been open-sourced and integrated into vLLM.

**Questions:**

1. How does lookup time, speedup, and scoring approximation (compared to best lookup) scale with the number of records? (within some distributional assumptions)
2. Can you explain the title?
3. 'Related Works' section is in a bit of a weird spot. Why not together with ‘Background’?  If you were trying to point out tradeoffs or highlight most recent works in particular, I suggest renaming the section.

**Ethical Concerns:**

["NO or VERY MINOR ethics concerns only"]

**Final Justification:**

The original review stands, and the decision is only strengthened by the degree to which the authors addressed each of my questions and concerns.

**Limitations:**

yes

**Paper Formatting Concerns:**

No major issues I saw.

**Quality:**

4

**Strengths And Weaknesses:**

## Strengths
* Paper is clear and well-written
* Motivation is clear, and emphasized through the mention of patterns that are particularly useful for the priors of this method: self-consistency, self-refinement, multi-agent workflows
* Method:
  * Simple: easy to understand, no added complexity that isn’t justified, model-free
  * Practical/general: doesn’t require strict homogeneity of LLMs within the multi-agent setting or training an auxiliary model; demonstrated use outside of multi-agent systems as well (with the caveat of reduced benefit in these settings)
  * Open-sourced integration into vLLM – good community contribution
  * Handles the tradeoff in number of consecutive draft tokens vs likelihood of them being incorrect in an adaptive rather than purely heuristic fashion, adapting specifically to the distribution driven by that agentic system.
  * Mentions practical additions, like maintaining two trees – one that’s global and one that’s smaller, per-request
  * Significant gains, including comparison to and performance against EAGLE
  * Admits to limitations compared to existing methods in certain settings

## Weaknesses
* The scoring/lookup is the key challenge to overcome and could have been highlighted/explained better
* The title is strange to me. 'Extreme' in the title seems a bit *extreme* -- I'd expect that word only with orders of magnitude difference.  I also didn't understand what 'Emerging' means here.  Are you referring to agents/structured outputs?
* Proprietary benchmarks will make it hard for the community to reproduce/compare
* Doesn't address scenarios when cpu memory is filled -- many practical scenarios at scale when this might happen.

---

> ### Author Rebuttal · Authors · 2025-07-31
>
> Thank you for your positive review and detailed feedback, our response is below.
>
> **Scalability of lookup time, speedup and scoring.**
>
> For lookup time vs number of records, we provide an additional micro-benchmark of the suffix tree below, where we measure its lookup time, update time, and memory consumption for various tree sizes. We will include it in our final paper.
>
> | Total \# entries | Total \# tokens | Memory footprint (MB) | Avg update time per token (μs) | Avg lookup time per speculated token (μs) |
> | :---- | :---- | :---- | :---- | :---- |
> | 1 K entries | 27 M | 1670 MB | 3.95 | 12.18 |
> | 5 K entries | 143 M | 2980 MB | 4.06 | 10.93 |
> | 10 K entries | 285 M | 4070 MB | 4.05 | 12.00 |
> | 15 K entries | 429 M | 5110 MB | 4.07 | 11.62 |
> | 20 K entries | 572 M | 6150 MB | 4.04 | 11.64 |
>
> In short, the total memory consumption scales linearly with the size of the tree, and both the update and lookup times remain fast even with larger trees.
>
> For speculative speedup and accuracy vs number of records, we provided a brief ablation study in Fig 8\. Speedup consistently improves as suffix tree size increases, demonstrating that more historical data leads to better performance. Importantly, acceptance rates remain stable (\~10-20%) even as tree size varies dramatically, indicating that our adaptive speculation mechanism (MAX\_SPEC \= αp) effectively manages the quality-quantity tradeoff.
>
> If there are specific questions beyond what’s provided in our ablation study, we are happy to answer during the discussion period.
>
> **Scenarios when CPU memory is filled up.**
>
> We agree that the CPU memory can be filled up, but we would like to work through a practical scenario. Our micro-benchmark results above show that Suffix Decoding is highly memory efficient, caching 572M tokens using only 6.15 GB of memory, and scaling linearly. At the same time, optimized serving systems can generate on the order of \~5000 tokens/s for Llama-8B on an A100 GPU \[1\]. Typical A100 systems (e.g. AWS p4d.24xlarge) have 144GB of CPU memory per A100 GPU. This means that Suffix Decoding can potentially cache 31 days of generated tokens per A100 GPU before hitting CPU memory limits (see math below). Beyond this limit, it is also straightforward to incorporate cache eviction (e.g. LRU) into Suffix Decoding to avoid out-of-memory problems.
>
> Time to fill up GPU memory:
>
> 1. Memory per token: $$\frac{6.15 GB}{572M tokens} = 1.075 \times 10^{-8} GB/token$$
> 2. Total token capacity: $$\frac{144 GB}{(1.075 \times 10^{-8} GB/token)} = 1.34 \times 10^{10} tokens$$
> 3. Time to fill RAM: $$\frac{1.34 \times 10^{10} tokens}{5000 tok/s} = 2.68 \times 10^{6} seconds =  \boxed{31 days}$$
>
>
> **Can you explain the title?**
>
> We used the term "extreme" in reference to the significantly longer speculation sequences possible with suffix trees compared to typical speculative decoding, which is also crucial for exploiting repetition in agentic applications such as SWE-Bench and AgenticSQL. We used the term "emerging" in reference to the growing field of agentic AI applications that involve multiple steps in answering a user query, often with highly repetitive LLM calls. We will add additional explanations and a footnote to our introduction to explain these choices.
>
> **'Related Works' section is in a bit of a weird spot.**
>
> We thank the reviewer for pointing out the non-standard location of the related works section and we will move it closer to the background section as suggested.
>
> \[1\] (Achieving Faster Open-Source Llama3 Serving with SGLang Runtime (vs. TensorRT-LLM, vLLM) | SGLang Team, Jul 25, 2024 | LMSYS Org)

---

> > ### Author Response · Authors · 2025-08-06
> >
> > We hope our response has sufficiently addressed your main concerns. Please let us know if there are any further clarifications, information, or updates to our paper that would impact your assessment and rating of our paper.

---

> > ### Comment · Reviewer_CbVA · 2025-08-07
> >
> > Thank you for addressing the questions, especially with the quantitative values backing up the practical issue of CPU memory.
> >
> > Thanks for explaining 'extreme' as well -- I think it's important to back up such a strong word in the title.
> >
> > Great work; looking forward to seeing it at NeurIPS.

---

### Official Review · Reviewer_WAC7 · 2025-07-03

**Clarity:** 2
**Significance:** 3
**Originality:** 3
**Rating:** 5
**Confidence:** 4

**Summary:**

The paper introduces SuffixDecoding, a model-free speculative decoding framework tailored for agentic LLM pipelines where generations exhibit high repetitiveness and structured loops. Instead of relying on a smaller draft model, SuffixDecoding constructs a suffix tree over past prompts and completions to identify the longest matching context at each decoding step. It then greedily expands a “speculation tree” of frequent continuations, verifies the selected prefix in a single forward pass of the full LLM, and adjusts its maximum speculation length dynamically based on match depth. Evaluated on two real-world agentic benchmarks (SWE-Bench, AgenticSQL) and in live vLLM, SuffixDecoding achieves up to 3.9× speedups on offline tasks and 4.5× end-to-end gains, while preserving exact token fidelity.

**Questions:**

1. Latency and profiling
- Please report real-world latency and energy consumption comparisons between SuffixDecoding and a standard draft-model speculative decoder on GPU. This will verify that FLOP savings translate into actual system-level speedups.

2. Memory overhead
- How large do the suffix trees grow in practice, and what are the RAM and update-time costs under concurrent agentic requests? Empirical profiling or a detailed complexity analysis would clarify deployment feasibility.

3. Fallback threshold
- The hybrid threshold τ governs when to abandon tree-based speculation. Can you characterize its sensitivity across mixed workloads? A small study varying τ on open-ended versus highly repetitive streams would help practitioners tune the system.

4. Generalization beyond Agentic workloads
- Have you evaluated SuffixDecoding on less structured or single-turn tasks (e.g., open-ended question answering)? Understanding its failure modes and identifying break-even points would guide broader application.

**Note: In particular, providing detailed latency and memory-overhead measurements across diverse tasks, along with a thorough analysis of τ’s sensitivity, will likely eliminate any remaining concerns and could substantially raise the paper’s overall score.**

**Ethical Concerns:**

["NO or VERY MINOR ethics concerns only"]

**Final Justification:**

Well structured clarification to my previous weakness and questions.

**Limitations:**

yes

**Quality:**

2

**Strengths And Weaknesses:**

### Quality
* The authors compare against both model-based speculators (EAGLE-2/3) and other model-free baselines (Token Recycling, Prompt Lookup), demonstrating consistent speed and throughput gains. – The paper reports FLOP-based efficiency but omits wall-clock latency and energy measurements on GPU hardware, leaving system-level impact unquantified.

### Clarity
* The algorithmic pipeline is described clearly, with pseudocode guiding the reader through tree construction, speculation, and verification steps. Figures illustrate key data structures and control flow. – Details around memory management (suffix-tree storage, updates under concurrency) and fallback threshold tuning (τ) are only sketched in prose, making reproduction nontrivial.

### Significance
* By decoupling speculative decoding from auxiliary draft models, SuffixDecoding opens a new direction for model-free inference acceleration in settings where prompt/response patterns repeat. – Its applicability to more open-ended or non-agentic workloads remains unclear: acceptance rates vary and no clear decision rule is given for when to revert to traditional draft-model approaches.

###Originality
* The use of suffix trees for speculative decoding is novel and exploits structural redundancies in agent-driven loops. – Prior work has explored cache-based and retrieval-augmented inference; the paper could more sharply delineate how suffix-tree speculation fundamentally differs in algorithmic guarantees or failure modes.

---

> ### Author Rebuttal · Authors · 2025-07-31
>
> Thank you for your detailed review and feedback, our response is below.
>
> **Latency and profiling (the paper reports FLOP-based efficiency but omits wall-clock latency and energy measurements on GPU hardware).**
>
> We believe there is a misunderstanding regarding the metrics reported in our paper. To clarify, all speedup measurements in our paper are based on wall-clock latency, not FLOP counts. The speedup in Fig 4 is the ratio of real-world wall-clock time per output token measured on H100 GPUs. For example, in SWE-Bench, Suffix Decoding achieves 31.093 ms per token vs vanilla's 49.836 ms per token (see Appendix A.1.2), resulting in a speedup of 49.836 / 31.093 \= 1.6x.
>
> FLOPs and energy consumption are less relevant for evaluating speculative decoding methods because speculative decoding typically expends *more* FLOPs (via speculation) to obtain *lower* wall-clock latency (via parallel verification).
>
> **Memory overhead: How large do the suffix trees grow in practice, and what are the RAM and update-time costs under concurrent agentic requests?**
>
> The suffix trees employed by Suffix Decoding are highly time and memory efficient. To illustrate this, we present a microbenchmark below showing the memory footprint and per-token update cost across various tree sizes (using requests from the SWE-Bench dataset). We will include this additional microbenchmark in the revised paper.
>
> | Total \# entries | Total \# tokens (millions) | CPU Memory (MB) | Avg update time per token over last 10 entries (μs) |
> | :---- | :---- | :---- | :---- |
> | 1 K entries | 27 M | 1670 MB | 3.95 |
> | 5 K entries | 143 M | 2980 MB | 4.06 |
> | 10 K entries | 285 M | 4070 MB | 4.05 |
> | 15 K entries | 429 M | 5110 MB | 4.07 |
> | 20 K entries | 572 M | 6150 MB | 4.04 |
>
> Overall, the memory consumption scales linearly with the size of the suffix tree, while the update time remains consistently low. For comparison, optimized inference engines running Llama-8B on an A100 GPU can typically generate \~5000 tokens per second \[1\] or \~432M tokens per day. Typical A100 systems (e.g. AWS p4d.24xlarge) have 144GB of CPU memory per A100 GPU. This means that Suffix Decoding can potentially cache 31 days of generated tokens per A100 GPU before hitting CPU memory limits. Beyond this limit, it is also straightforward to incorporate cache eviction (e.g. LRU) into Suffix Decoding to avoid out-of-memory problems.
>
>
>
> **Generalization beyond Agentic workloads: Have you evaluated SuffixDecoding on less structured or single-turn tasks (e.g., open-ended question answering)?**
>
> To clarify, the Spec-Bench dataset used in our evaluation (Sec 4.1, Fig 4\) consists of mainly non-agentic, open-ended tasks (e.g. single-turn question-answering). We included this dataset explicitly to stress-test Suffix Decoding in a scenario where *we did not expect it to perform well*, as evaluating performance under these extreme scenarios is important to understand the limitations and robustness of Suffix Decoding.
>
> The tables in appendix A.1.2 detailed performance results across all 13 Spec-Bench categories. For example, the `qa` (question answering) column represents open-ended, single-turn questions. Among the 13 categories, `qa`, `rag`, `math_reasoning`, `summarization`, and `translation` are single-turn tasks.
>
> Although Suffix Decoding is not specifically designed for these open-ended tasks, it can be combined with other speculative decoding methods in a *hybrid* mode to achieve the best of both worlds. We have performed additional experiments evaluating this hybrid-mode, which we present below.
>
> **SpecBench \- Speedup (x)**
>
> | System | coding | extraction | humanities | math | math\_reasoning | qa | rag | reasoning | roleplay | stem | summarization | translation | writing | Overall |
> | :---- | :---- | :---- | :---- | :---- | :---- | :---- | :---- | :---- | :---- | :---- | :---- | :---- | :---- | :---- |
> | hybrid (τ=7) | 3.441 | 2.795 | 2.800 | 3.139 | 2.866 | 2.404 | 2.542 | 2.496 | 2.461 | 2.871 | 2.573 | 1.759 | 2.833 | 2.500 |
> | eagle3 | 3.115 | 2.634 | 2.713 | 2.910 | 2.888 | 2.172 | 2.474 | 2.446 | 2.394 | 2.764 | 2.520 | 1.447 | 2.622 | 2.367 |
> | recycling | 2.619 | 2.234 | 2.209 | 2.660 | 2.577 | 1.951 | 2.057 | 2.150 | 1.979 | 2.334 | 2.218 | 1.932 | 2.057 | 2.169 |
> | eagle2 | 2.548 | 1.998 | 1.958 | 2.215 | 2.220 | 1.733 | 1.877 | 1.968 | 1.751 | 2.007 | 1.756 | 1.336 | 1.791 | 1.825 |
> | eagle | 2.427 | 1.956 | 1.848 | 2.142 | 2.112 | 1.600 | 1.841 | 1.871 | 1.703 | 1.922 | 1.702 | 1.305 | 1.753 | 1.752 |
> | suffix | 1.828 | 1.630 | 1.383 | 1.967 | 1.544 | 1.773 | 1.890 | 1.389 | 1.156 | 1.386 | 1.618 | 1.618 | 1.327 | 1.659 |
> | pld | 1.712 | 1.484 | 1.231 | 1.753 | 1.323 | 1.326 | 1.807 | 1.320 | 1.055 | 1.275 | 1.628 | 1.219 | 1.232 | 1.448 |
> | vanilla | 1.000 | 1.000 | 1.000 | 1.000 | 1.000 | 1.000 | 1.000 | 1.000 | 1.000 | 1.000 | 1.000 | 1.000 | 1.000 | 1.000 |
>
> **AgenticSQL \- Speedup (x)**
>
> | System | CATEGORIZATION | FEATURE\_EXTRACTION | QUESTION\_SUGGESTION | SQL\_COMBINE | SQL\_FANOUT1 | SQL\_FANOUT2 | SQL\_FANOUT3 | Overall |
> | :---- | :---- | :---- | :---- | :---- | :---- | :---- | :---- | :---- |
> | suffix | 3.016 | 9.854 | 10.406 | 4.848 | 3.205 | 2.839 | 3.211 | 5.345 |
> | hybrid (τ=1) | 2.309 | 7.025 | 8.368 | 3.382 | 2.024 | 2.421 | 2.072 | 3.947 |
> | recycling | 2.588 | 3.472 | 2.672 | 2.640 | 2.502 | 2.604 | 2.492 | 2.710 |
> | pld | 1.330 | 3.695 | 1.255 | 3.298 | 1.936 | 1.311 | 1.905 | 2.105 |
> | eagle2 | 1.591 | 2.751 | 1.673 | 1.855 | 1.527 | 2.119 | 1.527 | 1.864 |
> | eagle3 | 1.328 | 1.720 | 2.025 | 1.619 | 1.108 | 2.496 | 1.056 | 1.623 |
> | eagle | 1.324 | 2.246 | 1.598 | 1.669 | 1.229 | 1.955 | 1.138 | 1.595 |
> | vanilla | 1.000 | 1.000 | 1.000 | 1.000 | 1.000 | 1.000 | 1.000 | 1.000 |
>
> Here, we used a fallback threshold of τ=7 for SpecBench, and τ=1 for AgenticSQL. For more open-ended tasks such as QA (question answering), the hybrid approach with Suffix Decoding outperforms the speculative decoding method alone. This indicates that *while Suffix Decoding can greatly improve agentic applications, it does not have to come at the expense of other open-ended applications*.
>
> Note that all baselines are slightly improved compared to the table in our submission due to performance improvements in the benchmark framework.
>
> Given space limitations, we are presenting only the wall-clock speedups compared to the baseline. A comprehensive breakdown of results, including per-token latency (TPOT), mean accepted tokens per step (MAT), mean acceptance rate, and speculation time per generated token, will be incorporated into the revised paper.
>
> **Hybrid Fallback Threshold: Can you characterize its sensitivity across mixed workloads?**
>
> We report the results of SpecBench and AgenticSQL experiments using the Hybrid Suffix Decoding method with various thresholds. In the table below, we show the measured wall-clock speedups with respect to the vanilla (no speculation) baseline.
>
> **SpecBench \- Speedup (x)**
>
> | System | coding | extraction | humanities | math | math\_reasoning | qa | rag | reasoning | roleplay | stem | summarization | translation | writing | Overall |
> | :---- | :---- | :---- | :---- | :---- | :---- | :---- | :---- | :---- | :---- | :---- | :---- | :---- | :---- | :---- |
> | τ=7 | 3.441 | 2.795 | 2.800 | 3.139 | 2.866 | 2.404 | 2.542 | 2.496 | 2.461 | 2.871 | 2.573 | 1.759 | 2.833 | 2.500 |
> | τ=5 | 3.247 | 2.701 | 2.682 | 2.702 | 2.605 | 2.280 | 2.402 | 2.351 | 2.358 | 2.735 | 2.461 | 1.763 | 2.697 | 2.366 |
> | τ=6 | 3.186 | 2.595 | 2.614 | 2.875 | 2.567 | 2.216 | 2.360 | 2.296 | 2.304 | 2.675 | 2.380 | 1.719 | 2.610 | 2.314 |
> | τ=4 | 3.072 | 2.561 | 2.552 | 2.719 | 2.402 | 2.219 | 2.278 | 2.249 | 2.276 | 2.569 | 2.285 | 1.728 | 2.641 | 2.249 |
> | τ=3 | 2.875 | 2.407 | 2.352 | 2.536 | 2.195 | 2.086 | 2.166 | 2.099 | 2.161 | 2.387 | 2.161 | 1.733 | 2.503 | 2.126 |
> | τ=2 | 2.734 | 2.256 | 2.218 | 2.401 | 1.980 | 2.041 | 2.061 | 2.011 | 1.990 | 2.194 | 2.026 | 1.791 | 2.362 | 2.028 |
> | τ=1 | 2.284 | 1.977 | 1.935 | 2.014 | 1.620 | 1.858 | 1.870 | 1.694 | 1.745 | 1.897 | 1.786 | 1.723 | 2.117 | 1.802 |
> | τ=0 (suffix alone) | 1.828 | 1.630 | 1.383 | 1.967 | 1.544 | 1.773 | 1.890 | 1.389 | 1.156 | 1.386 | 1.618 | 1.618 | 1.327 | 1.659 |
>
> **AgenticSQL \- Speedup (x)**
>
> | System | CATEGORIZATION | FEATURE\_EXTRACTION | QUESTION\_SUGGESTION | SQL\_COMBINE | SQL\_FANOUT1 | SQL\_FANOUT2 | SQL\_FANOUT3 | Overall |
> | :---- | :---- | :---- | :---- | :---- | :---- | :---- | :---- | :---- |
> | τ=0 (suffix alone) | 3.016 | 9.854 | 10.406 | 4.848 | 3.205 | 2.839 | 3.211 | 5.345 |
> | τ=1 | 2.309 | 7.025 | 8.368 | 3.382 | 2.024 | 2.421 | 2.072 | 3.947 |
> | τ=2 | 2.243 | 7.137 | 7.965 | 3.327 | 1.993 | 2.483 | 1.405 | 3.799 |
> | τ=7 | 2.059 | 6.794 | 7.096 | 3.440 | 1.692 | 2.812 | 1.714 | 3.663 |
> | τ=8 | 2.080 | 6.848 | 4.959 | 3.358 | 1.691 | 2.854 | 1.259 | 3.298 |
> | τ=3 | 2.217 | 5.505 | 5.679 | 3.517 | 1.913 | 2.256 | 1.424 | 3.220 |
> | τ=4 | 2.173 | 5.669 | 5.404 | 3.526 | 1.892 | 2.237 | 1.390 | 3.189 |
> | τ=5 | 2.023 | 5.211 | 5.139 | 3.575 | 1.868 | 2.376 | 1.326 | 3.078 |
> | τ=6 | 2.027 | 5.113 | 5.199 | 3.491 | 1.782 | 2.351 | 1.287 | 3.040 |
>
> For open-ended generation, a good heuristic is to set the fallback threshold (τ) to a value close to or slightly exceeding the mean number of accepted tokens (MAT) for the model-based speculator used with the suffix-tree. For instance, we used EAGLE-3, which has a MAT of approximately 4.65 tokens/step in SpecBench. The table shows that τ values of 5, 6, and 7 yield the best overall speedups, with the results being quite similar across these values.
>
> Conversely, in agentic workloads like AgenticSQL, the standalone SuffixDecoding implementation (τ=0) is optimal because it can confidently speculate much longer sequences than model-based speculators such as EAGLE-3.
>
> \[1\] (Achieving Faster Open-Source Llama3 Serving with SGLang Runtime (vs. TensorRT-LLM, vLLM) | SGLang Team, Jul 25, 2024 | LMSYS Org)

---

> ### Comment · Reviewer_WAC7 · 2025-08-06
>
> I’m happy to raise my score in light of these clarifications.
>
> Separately, the paper would benefit from citing prior work on independent multi-token heads in its Related Works, for example [1,2,3]:
>
> [1] Better & Faster LLMs via Multi-Token Prediction, Gloeckle et. al., ICML 24.
>
> [2] Accelerating Blockwise Parallel Language Models with Draft Refinement, Kim et. al, NeurIPS 24.
>
> [3] Blockwise Parallel Decoding for Deep Autoregressive Models, Stern et. al, NIPS 18.

---

> > ### Author Response · Authors · 2025-08-06
> >
> > Thank you for reviewing our response and raising your score. We shall cite the additional works provided in the updated version of our paper.

---

### Note · Authors · 2025-08-12

We thank all reviewers for their constructive feedback. We are pleased that our responses have addressed their concerns, with Reviewer WAC7 raising their score and all reviewers providing positive acknowledgments.

---

## Addressing Reviewers' Main Concerns:

**Wall-clock latency measurements (Reviewer WAC7):** We clarified that ALL reported speedups are based on real wall-clock latency, not FLOPs. Our measurements on H100 GPUs show concrete time savings (e.g., 31.093ms vs 49.836ms per token on SWE-Bench).

**Memory overhead and scalability (Reviewers WAC7, CbVA, Bnmw):** We provided detailed microbenchmarks showing SuffixDecoding uses only 6.15GB for 572M tokens with consistent \~4μs update times. Given typical server configurations (144GB CPU RAM per A100), the system can cache \~31 days of generation before requiring eviction.

**Performance on non-agentic tasks (Reviewers WAC7, QNpe, Bnmw):** We clarified that the SpecBench dataset we evaluated contains primarily open-ended non-agentic tasks. We provided additional experiments that demonstrated hybrid SuffixDecoding+EAGLE-3 achieves 2.5x speedup on these tasks, outperforming EAGLE-3 alone (2.37x). This addresses concerns about generalization beyond agentic workloads and shows that SuffixDecoding can be complementary to existing speculative decoding methods

**Fallback threshold sensitivity (Reviewer WAC7):** We provided comprehensive experiments varying τ from 0-8, showing the system is robust with easily-determined optimal values (τ=5-7 for SpecBench, τ=0-1 for AgenticSQL).

**Title justification (Reviewer CbVA):** We clarified that "extreme" refers to the significantly longer speculation sequences (10+ tokens) compared to typical methods, while "emerging" refers to the growing field of agentic AI applications.

**Integration with online serving (Reviewer QNpe):** We explained how SuffixDecoding can integrate with existing batch optimization methods like TurboSpec, which dynamically adjusts speculation based on batch size and goodput metrics.

---

All reviewers acknowledged our responses adequately addressed their concerns. The feedback from reviewers helps us substantially improve our paper, and we will incorporate all suggested improvements in the final version.

---

### Decision · Program_Chairs · 2025-09-17

**Decision:**

Accept (spotlight)

**Comment:**

(a) Summary:
This paper proposes SuffixDecoding, a training-free speculative decoding method based on suffix tree. The method leverages the high repetitiveness and structured loops in agentic LLMs to achieve acceleration. Experiments on SWE-Bench and AgenticSQL show the power of the proposed method.

(b) Strengths:
1. Reviewers recognize the novelty in this paper.

2. The paper is well-written.

3. Experiments on agentic tasks show the advantages of the proposed method.

4. Training-free method is easier to be applied to more models with lower cost.

(c) Weaknesses:
1. The applicability of SuffixDecoding to more open-ended or non-agentic workloads remains unclear. Acceptance rates vary for different dataset. Multiple reviewers have concerns that for non-agentic tasks and non-repetitive workloads, the proposed method might lead to limited speedup.

2. How suffix-tree speculation fundamentally differs from cache-based and retrieval-augmented inference in algorithmic guarantees or failure modes remains unclear.

3. Some references are missing.

4. The reported numbers of baseline methods seem to be worse than the reported numbers in the original papers. It is known that the choice of hyper-parameters matters. It is unclear how this paper tunes the hyper-parameters for baseline methods (e.g., tree width, tree depth, tree size, etc when using tree attention).

(d) Why the decision:
Overall, the paper is of good quality. All reviewers vote for accept. AC stands with the reviewers and recommend accept with spotlight.

(e) Summary of discussions:
In the first round of review, all reviewers are positive about the paper. The rebuttal further strengthens the advantage of this paper. AC would recommend accept too.